# Altered bone growth dynamics prefigure craniosynostosis in a zebrafish model of Saethre-Chotzen syndrome

Camilla S Teng[1,2], Man-chun Ting[2], D'Juan T Farmer[1], Mia Brockop[2], Robert E Maxson[2]*, J Gage Crump[1]*

[1]Department of Stem Cell Biology and Regenerative Medicine, University of Southern California, Los Angeles, United States; [2]Department of Biochemistry, Keck School of Medicine, University of Southern California, Los Angeles, United States

**Abstract** Cranial sutures separate the skull bones and house stem cells for bone growth and repair. In Saethre-Chotzen syndrome, mutations in *TCF12* or *TWIST1* ablate a specific suture, the coronal. This suture forms at a neural-crest/mesoderm interface in mammals and a mesoderm/mesoderm interface in zebrafish. Despite this difference, we show that combinatorial loss of *TCF12* and *TWIST1* homologs in zebrafish also results in specific loss of the coronal suture. Sequential bone staining reveals an initial, directional acceleration of bone production in the mutant skull, with subsequent localized stalling of bone growth prefiguring coronal suture loss. Mouse genetics further reveal requirements for *Twist1* and *Tcf12* in both the frontal and parietal bones for suture patency, and to maintain putative progenitors in the coronal region. These findings reveal conservation of coronal suture formation despite evolutionary shifts in embryonic origins, and suggest that the coronal suture might be especially susceptible to imbalances in progenitor maintenance and osteoblast differentiation.
DOI: https://doi.org/10.7554/eLife.37024.001

*For correspondence:
maxson@usc.edu (REM);
gcrump@usc.edu (JGC)

**Competing interests:** The authors declare that no competing interests exist.

## Introduction

Craniofacial anomalies are among the most common congenital defects, encompassing cleft lip and palate, facial malformations, and abnormalities in the flat bones forming the top of the skull. Craniosynostosis involves the premature fusion of the skull bones at sutures, fibrous structures that join the bones. During normal development, cranial sutures hold bones of the skull in place while providing malleability required during childbirth. Cranial sutures also contain stem cells that allow for continued skull bone growth as the brain and head structure expands (*Zhao et al., 2015*). While human skull bones eventually fuse later in life, precocious bone fusion in craniosynostosis correlates with region-specific defects in bone growth that often negatively impact growth of the underlying brain. Genetic causes of craniosynostosis include mutations in genes that participate in developmental signaling pathways, including *FGFR*s *1*, *2*, and *3* (*Hajihosseini, 2008*), *TGFBR*s *1* and *2* (*Loeys et al., 2005*), and the Notch ligand *JAGGED1* (*Kamath et al., 2002*). A central unanswered question is the extent to which defects in these pathways result in a failure to maintain postnatal stem cells once the sutures have formed, or whether these pathways have roles in the specification and maintenance of bone progenitors during earlier phases of bone growth preceding suture formation (*Flaherty et al., 2016*). Most studies have focused on the maintenance of postnatal sutural stem cells, as these stem cells have only been recently marked with a variety of reporters in mice, for example based on *Gli1* (*Zhao et al., 2015*), *Axin2* (*Maruyama et al., 2016*), and *Prrx1* (*Wilk et al., 2017*). Less is known, however, about how potential defects in the embryonic progenitors that grow the skull bones may prefigure suture loss, although a few reports describe altered bone formation during embryonic

**eLife digest** Some of the most common birth defects involve improper development of the head and face. One such birth defect is called craniosynostosis. Normally, an infant's skull bones are not fully fused together. Instead, they are held together by soft tissue that allows the baby's skull to more easily pass through the birth canal. This tissue also houses specialized cells called stem cells that allow the brain and skull to grow with the child. But in craniosynostosis these stem cells behave abnormally, which fuses the skull bones together and prevents the skull and brain from growing properly during childhood.

One form of craniosynostosis called Saethre-Chotzen syndrome is caused by mutations in one of two genes that ensure the proper separation of two bones in the roof of the skull. Mice with mutations in the mouse versions of these genes develop the same problem and are used to study this condition. Mouse studies have looked mostly at what happens after birth. Studies looking at what happens in embryos with these mutations could help scientists learn more. One way to do so would be to genetically engineer zebrafish with the equivalent mutations. This is because zebrafish embryos are transparent and grow outside their mother's body, making it easier for scientists to watch them develop.

Now, Teng et al. have grown zebrafish with mutations in the zebrafish versions of the genes that cause Saethre-Chotzen syndrome. In the experiments, imaging tools were used to observe the live fish as they developed. This showed that the stem cells in their skulls become abnormal much earlier than previous studies had suggested. Teng et al. also showed that similar stem cells are responsible for growth of the skull in zebrafish and mice.

Babies with craniosynostosis often need multiple, risky surgeries to separate their skull bones and allow their brain and head to grow. Unfortunately, these bones often fuse again because they have abnormal stem cells. Teng et al. provide new information on what goes wrong in these stem cells. Hopefully, this new information will help scientists to one day correct the defective stem cells in babies with craniosynostosis, thus reducing the number of surgeries needed to correct the problem.
DOI: https://doi.org/10.7554/eLife.37024.002

stages in craniosynostosis mutants (*Merrill et al., 2006*). The ex utero development and transparency of zebrafish provide a unique opportunity to track the progenitors that form the skull bones, and to better understand how defects in the dynamics of bone growth relate to a later ability to form and maintain sutures once the bones come together (*Quarto and Longaker, 2005*; *Laue et al., 2011*; *Kague et al., 2016*; *Topczewska et al., 2016*).

A striking feature of many forms of craniosynostosis is that only particular sutures are affected. For example, in Saethre-Chotzen syndrome, the second most common form of craniosynostosis, the coronal suture is selectively lost. The majority of Saethre-Chotzen patients harbor heterozygous loss-of-function mutations in *TWIST1* or *TCF12*, which encode basic helix-loop-helix transcription factors (*el Ghouzzi et al., 1997*; *Howard et al., 1997*; *Sharma et al., 2013*). Similarly, mice lacking one copy of *Twist1*, or compound heterozygous for *Twist1* and *Tcf12*, display loss of just the coronal suture (*Sharma et al., 2013*). Analysis of mouse models has led to an appreciation of proper cell migration (*Yoshida et al., 2008*; *Ting et al., 2009*; *Roybal et al., 2010*; *Deckelbaum et al., 2012*), segregation of osteogenic and non-osteogenic cells at the sutural boundary (*Merrill et al., 2006*; *Ting et al., 2009*; *Yen et al., 2010*), and maintenance of postnatal stem cells (*Zhao et al., 2015*) in suture patency. However, it remains unknown the extent to which defects in early progenitors versus postnatal stem cells account for suture loss. As fish are amenable to repeated live imaging of calvarial bone growth and suture development outside the mother, this model provides an opportunity to correlate early defects in osteoprogenitors and bone growth with later suture loss in individual mutants, particularly in cases where the synostosis phenotype is variably penetrant. However, a potential complication is that the coronal suture of zebrafish (*Kague et al., 2012*; *Mongera et al., 2013*), as well as those of some amphibians (*Piekarski et al., 2014*) and birds (*Matsuoka et al., 2005*), forms at a mesoderm/mesoderm boundary, contrasting with the mammalian coronal suture that forms at a unique interface between the neural-crest-derived frontal bone and the mesoderm-derived parietal bone (*Ishii et al., 2015*) (*Figure 1A*). Mammalian sutures may therefore not be

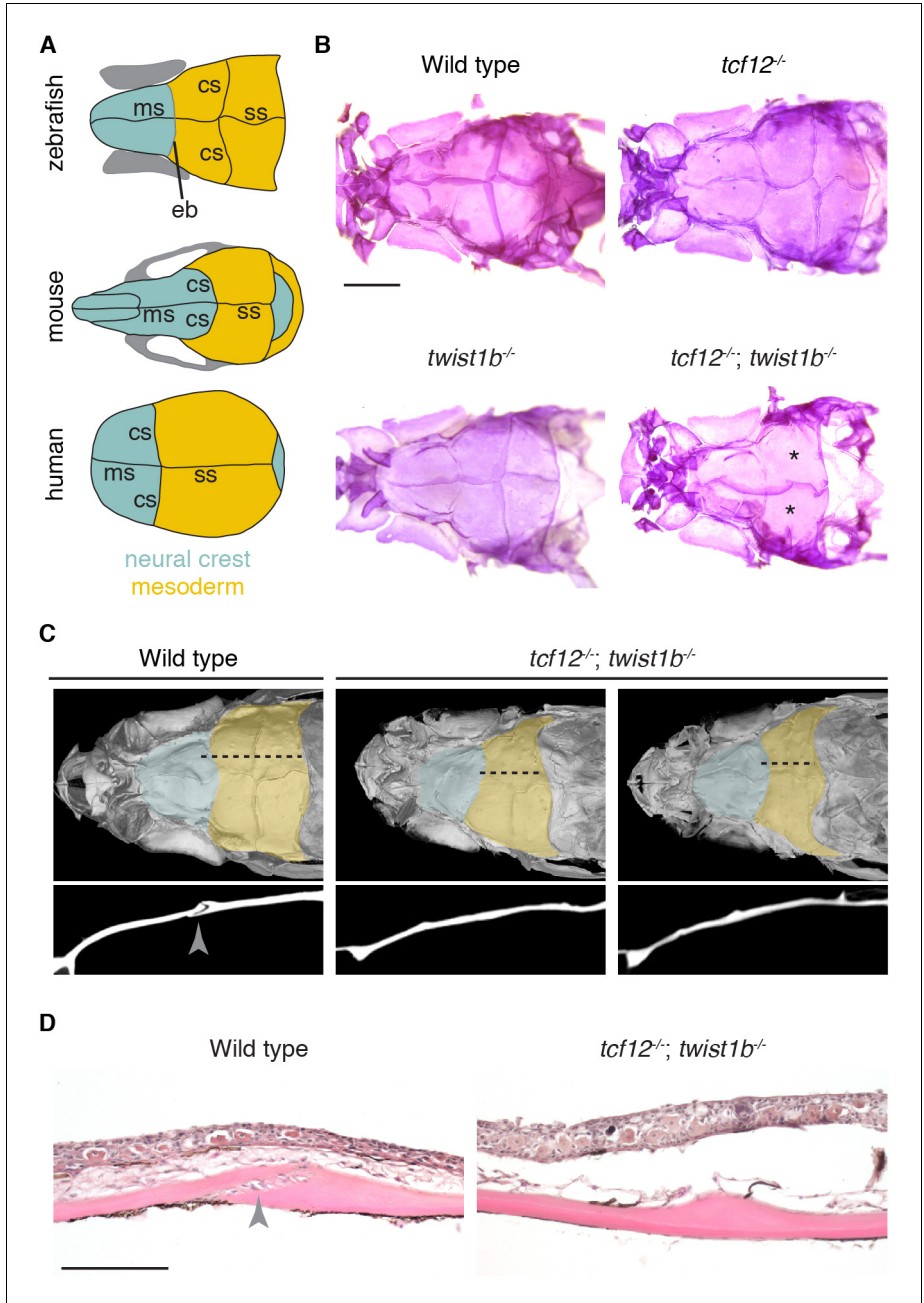

**Figure 1.** Coronal suture loss in *tcf12; twist1b* mutant zebrafish. (A) Diagrams of zebrafish, mouse, and human skulls, with neural crest contributions in turquoise and mesoderm contributions in gold. The coronal suture is at a mesoderm-mesoderm boundary in zebrafish and a neural-crest-mesoderm boundary in mouse and human. Instead of a suture, an epiphyseal bar cartilage (eb) is present at the neural-crest-mesoderm boundary in zebrafish. ms, metopic suture; ss, sagittal suture. (B) Dissected skullcaps of adult fish stained with Alizarin Red show loss of the coronal suture (asterisks) in *tcf12*[-/-]; *twist1b*[-/-] double mutants but not single mutants. Scale bar, 1 mm. (C) Micro-CT scans of adult fish heads show unilateral (left) and bilateral (right) coronal suture loss in *tcf12*[-/-]; *twist1b*[-/-] mutants. Shading indicates bone derived from neural crest (turquoise) and mesoderm (gold). Panels below are digital sections through the coronal sutures indicated by the dotted lines above. Arrowhead indicates the wild-type suture. (D) Hematoxylin and eosin-stained sections show loss of the coronal suture mesenchyme (arrowhead) in *tcf12*[-/-]; *twist1b*[-/-] mutants. Scale bar, 100 μm.

DOI: https://doi.org/10.7554/eLife.37024.003

The following figure supplements are available for figure 1:

**Figure supplement 1.** Zebrafish TALEN mutants.

*Figure 1 continued on next page*

*Figure 1 continued*

DOI: https://doi.org/10.7554/eLife.37024.004

**Figure supplement 2.** Patent sutures in *tcf12* single mutants.

DOI: https://doi.org/10.7554/eLife.37024.005

**Figure supplement 3.** Zebrafish skull phenotypes in different mutant combinations.

DOI: https://doi.org/10.7554/eLife.37024.006

considered evolutionarily homologous to the sutures of these other vertebrates, at least from a strict embryological perspective. Should the neural-crest/mesoderm boundary be a factor in coronal sensitivity, non-mammalian coronal sutures might not be susceptible to the same genetic perturbations that cause coronal-specific synostosis in mammals.

We report in this study the generation of a zebrafish model of Saethre-Chotzen syndrome that faithfully recapitulates the craniosynostosis phenotype seen in mice and humans with heterozygous mutations in *TCF12* and *TWIST1*. The similarity in the genetic interaction between Twist1 and Tcf12 in mice, humans, and fish, despite differences in the cell lineages that give rise to the bones, suggests that the underlying processes of coronal suture development and craniosynostosis are deeply conserved. We demonstrate that in *tcf12; twist1b* mutant fish, the frontal and parietal bones grow abnormally. In mutants, skull bones initiate normally, yet early bone growth is accelerated across the skull. However, subsequent bone growth selectively stalls at the future coronal suture that is destined to fuse. Moreover, sequential live imaging of individual mutant fish shows that the degree of later bone stalling predicts which animals will lose the coronal suture. We observe a similar misregulation of bone growth in $Tcf12^{+/-}; Twist1^{+/-}$ mutant mice, with tissue-specific removal of *Twist1* resulting in selective overgrowth of the frontal or parietal bones. Further, *Twist1* function must be perturbed within both neural crest- and mesoderm-derived bones, and not just the mesoderm-derived postnatal sutural mesenchyme, to prevent suture formation. At least in mice, we find that these altered bone growth dynamics may be due to changes in osteoblast proliferation and eventual depletion of $Gli1^+$ and $Gremlin1^+$ putative bone progenitors prior to the fusion of the bones. These findings demonstrate that Tcf12 and Twist1 have a conserved early function during skull bone growth to regulate the sustained production of osteoblasts, possibly through maintenance of osteoprogenitors, and that the coronal suture is particularly sensitive to defects in this process.

## Results

### Specific loss of the coronal suture in *tcf12; twist1b* mutant zebrafish

In order to investigate requirements for Tcf12 and Twist1 homologs in zebrafish suture formation, we designed TALE nucleases to generate mutant alleles for *tcf12* and both zebrafish Twist1 homologs, *twist1a* and *twist1b*. The $tcf12^{el548}$, $twist1a^{el570}$, and $twist1b^{el571}$ alleles result in premature truncations before the helix-loop-helix domains required for DNA-binding and dimerization, thus likely abrogating all protein function (*Figure 1—figure supplement 1*). Whereas individual homozygous mutants displayed no gross defects as embryos or adults and had patent sutures across the head (*Figure 1B* and *Figure 1—figure supplement 2*), 38% of *tcf12; twist1b* double mutant adults developed unilateral or bilateral coronal synostosis, as revealed by Alizarin Red staining of bone (*Figure 1B*). We confirmed loss of the coronal suture by micro-computed tomography scans and histology (*Figure 1C,D*). As in mice and humans, coronal suture loss correlated with reduced anterior-posterior growth of the frontal and parietal bones, and in cases where the suture was lost unilaterally we consistently observed reduced anterior-posterior growth on that side of the skull (*Figure 1C*). In contrast, we did not detect a requirement for *twist1a* in suture development; sutures formed normally in $tcf12^{-/-}; twist1a^{-/-}$ mutants, and loss of *twist1a* alleles did not increase the severity or penetrance of suture defects in $tcf12^{-/-}; twist1b^{-/-}$ fish (*Figure 1—figure supplement 3*, *Supplementary file 1*). Loss of *tcf12* appears essential for synostosis, as rare adult viable $twist1a^{-/-}; twist1b^{-/-}$ fish had normal sutures. While we only detected fusions of the coronal suture in $tcf12^{-/-}; twist1b^{-/-}$ fish, we did observe other abnormalities in skull bones in different mutant combinations, including ectopic sutures and small gaps between the parietal bones (*Figure 1—figure supplement 3*, *Supplementary file 1*). Our results demonstrate that, as in humans and mice with reduced *TCF12*

and/or *TWIST1* dosage, mutations in the homologous genes in zebrafish primarily result in loss of the coronal suture, although other calvarial defects are also rarely observed.

## Loss of *tcf12* partially suppresses the embryonic defects and lethality of Twist1 deficiency

Given the synergistic effect of *tcf12* and *twist1b* loss on coronal suture formation, we examined whether *tcf12* also interacts genetically with Twist1 genes in earlier craniofacial development. Similar to previous reports of zebrafish with antisense morpholino reduction of *twist1a* and *twist1b* (*Das and Crump, 2012*), and mice with neural-crest-specific deletion of *Twist1* (*Bildsoe et al., 2009*), *twist1a*$^{-/-}$; *twist1b*$^{-/-}$ zebrafish embryos displayed defects in the specification of skeletogenic ectomesenchyme from the neural crest. In wild-type embryos, neural crest expression of *sox10* is down-regulated by 20 hr post-fertilization (hpf) as ectomesenchyme neural crest cells populate the pharyngeal arches. In *twist1a*$^{-/-}$; *twist1b*$^{-/-}$ embryos, we observed persistent *sox10* expression in arch ectomesenchyme and reductions in facial cartilage and bone at 5 days post-fertilization (dpf) (*Figure 2A,B*). Dorsal facial cartilages (e.g. palatoquadrate and hyosymplectic) were most affected (*Figure 2C*, *Figure 2—source data 1*), potentially reflecting the greater sensitivity of these elements to general neural crest defects (*Cox et al., 2012*) and/or post-migratory roles of Twist1 genes in branchial arch development (*Askary et al., 2017*). Interestingly, loss of *tcf12* suppressed rather than enhanced the severity of facial skeletal defects in *twist1a*$^{-/-}$; *twist1b*$^{-/-}$ larvae, with partial suppression seen with loss of just one *tcf12* allele (*Figure 2B,D*, *Figure 2—source data 2*). The suppression did not appear to be due to a rescue of ectomesenchyme specification, as similarly persistent *sox10* was evident in *twist1a*$^{-/-}$; *twist1b*$^{-/-}$ embryos with or without *tcf12* loss (*Figure 2A*). We also observed that loss of at least one copy of *tcf12* enhanced adult viability of *twist1a*$^{-/-}$; *twist1b*$^{-/-}$ mutants (*Figure 2E*, *Figure 2—source data 3*). These findings indicate temporally distinct genetic interactions between Twist1 and Tcf12, with Tcf12 acting antagonistically to Twist1 during arch development and synergistically during later skull bone growth and suture formation.

## Expression of *tcf12* and *twist1b* at multiple sutures in zebrafish

Given the selective loss of the coronal suture in mutants, we examined whether *tcf12* and *twist1b* genes might be selectively expressed in this suture. However, at a stage when sutures have just formed (14 mm standard length), we observed expression of *tcf12* and *twist1b* within the mesenchyme of not only the coronal but also the metopic and sagittal sutures (*Figure 3*). Whereas *twist1b* expression was largely restricted to the suture mesenchyme, *tcf12* expression was observed more broadly in the suture mesenchyme and cells surrounding the skull bones. Expression of *tcf12* and *twist1b* at multiple sutures in fish is consistent with similarly broad suture expression of *Twist1* in mice (*Rice et al., 2000*) and argues against genetic sensitivity of the coronal suture being due to selective expression of *tcf12* and *twist1b* at this suture.

## Altered calvarial bone growth prefigures coronal suture loss in *tcf12; twist1b* mutant fish

We next investigated whether suture defects in *tcf12; twist1b* mutants might result from an earlier misregulation of skull bone growth, to which the coronal suture might be particularly sensitive. In wild-type zebrafish, the anlage of the frontal and parietal bones can first be seen at 6 mm standard length, with the coronal suture forming at the interface of these bones in a lateral to medial progression (*Kague et al., 2016*). Live mineralization stains revealed that initiation of the frontal and parietal bones was unaffected in mutants, yet accelerated growth of these bones became detectable in mutants by 8 mm, and more so by 9 mm (*Figure 4A*). Staining with the mineralization dye Calcein Green revealed accelerated frontal bone fronts by 10.25 ± 0.25 mm in mutants, with no difference in the degree of increased bone between sides that developed synostosis and those that did not (*Figure 4B,C*, *Figure 4—source data 1*). Although the overall area of the mutant parietal bone was comparable to that of wild types at 10.25 ± 0.25 mm, both the mutant frontal and parietal bones were aberrantly shaped. In particular, both the frontal and parietal bones exhibited enhanced growth along an axis diagonal to the anterior-posterior and medial-lateral axes, which themselves exhibit little to no changes in directional growth (*Figure 4C*, *Figure 4—source data 2*). Such enhanced diagonal growth would be expected to bring the frontal and parietal bones closer

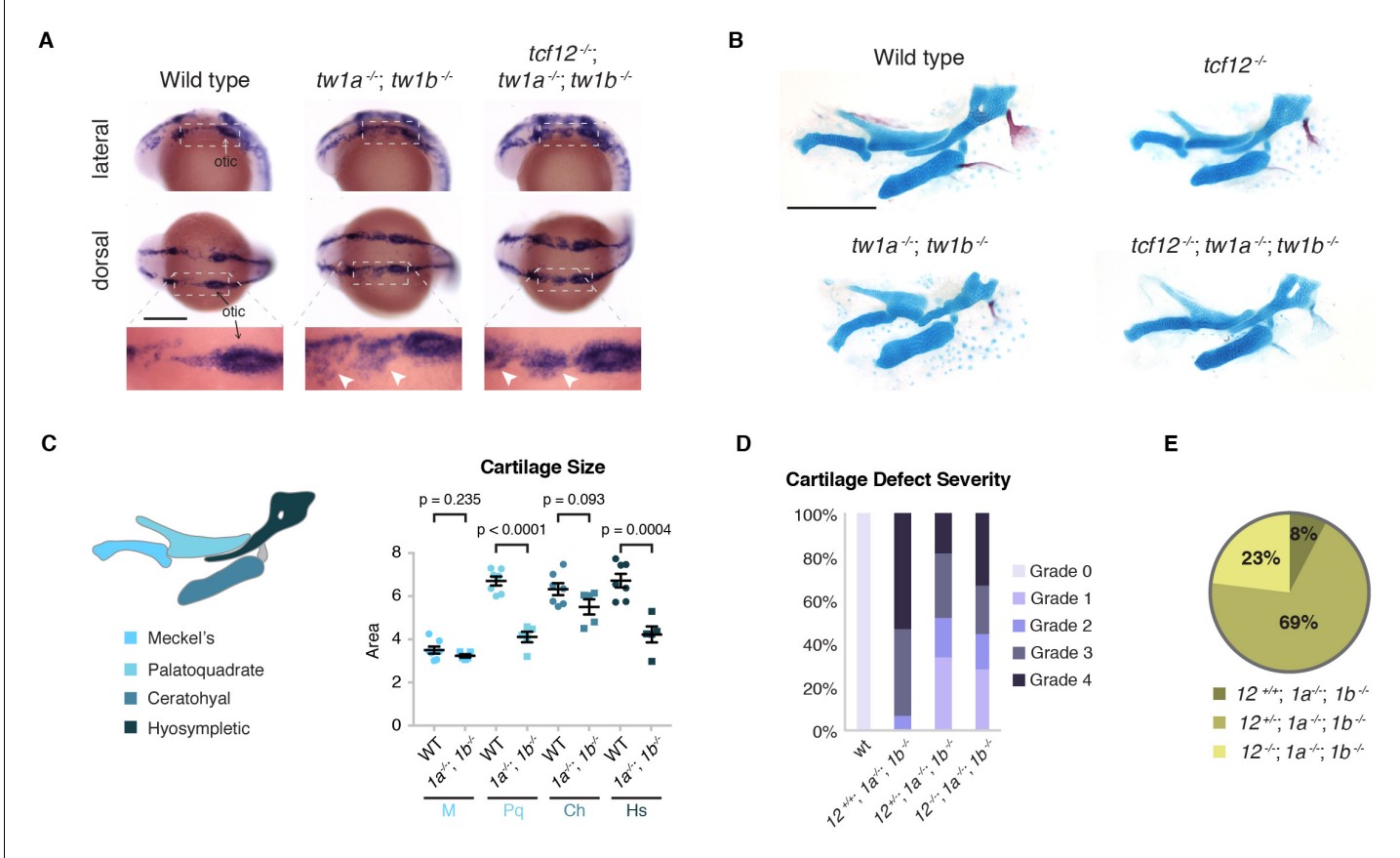

**Figure 2.** Loss of *tcf12* partially rescues *twist1a; twist1b* jaw cartilage defects and viability. (A) In situ hybridizations at 20 hpf show abnormal persistence of *sox10* expression in arch ectomesenchyme (boxed region) in *twist1a⁻/⁻; twist1b⁻/⁻* and *tcf12⁻/⁻; twist1a⁻/⁻; twist1b⁻/⁻* mutants. Arrowheads indicate persistent *sox10* expression in arches. (B) Unilateral dissections of the first and second arch skeletons stained with Alcian Blue (cartilage) and Alizarin Red (bone) at 5 dpf. Compared to the reductions of the upper facial skeleton in *twist1a⁻/⁻; twist1b⁻/⁻* mutants, *tcf12⁻/⁻; twist1a⁻/⁻; twist1b⁻/⁻* triple mutants display less severe defects. Scale bars, 250 μm. (C) Quantitation of wild-type and *twist1a⁻/⁻; twist1b⁻/⁻* jaw cartilage areas show specific reductions in more dorsal cartilages, the palatoquadrate (Pq) and hyosymplectic (Hs). M, Meckel's cartilage; Ch, ceratohyal. (D) Qualitative scoring of facial skeletal defects from Grade 0 (unaffected) to Grade 4 (most affected). Loss of one or two copies of *tcf12* improved the facial skeletal morphology of *twist1a⁻/⁻; twist1b⁻/⁻* mutants. Wild type (wt, n = 20), *twist1a⁻/⁻; twist1b⁻/⁻* (12⁺/⁺, 1a⁻/⁻, 1b⁻/⁻, n = 25), *tcf12⁺/⁻; twist1a⁻/⁻; twist1b⁻/⁻* (12⁺/⁻, 1a⁻/⁻, 1b⁻/⁻, n = 32), *tcf12⁻/⁻; twist1a⁻/⁻; twist1b⁻/⁻* (12⁻/⁻, 1a⁻/⁻, 1b⁻/⁻, n = 22). Using a Fisher's Exact Test, p=0.0032 for 12⁺/⁺, 1a⁻/⁻, 1b⁻/⁻ versus 12⁻/⁻, 1a⁻/⁻, 1b⁻/⁻ and p=0.001 for 12⁺/⁺, 1a⁻/⁻, 1b⁻/⁻ versus 12⁺/⁻, 1a⁻/⁻, 1b⁻/⁻. (E) Reduction of *tcf12* dosage improves adult viability of *twist1a⁻/⁻; twist1b⁻/⁻* mutants. From an incross of *tcf12⁺/⁻; twist1a⁺/⁻; twist1b⁺/⁻* fish, we obtained *twist1a⁻/⁻; twist1b⁻/⁻* mutants and assessed their viability to 3 months of age. After genotyping for *tcf12*, we observed a 4:36:12 ratio of *tcf12⁺/⁺*: *tcf12⁺/⁻*: *tcf12⁻/⁻*, compared to the predicted 13:26:13 ratio, which was significantly skewed as determined by a Chi-squared test (p=0.0062).

DOI: https://doi.org/10.7554/eLife.37024.007

The following source data is available for figure 2:

**Source data 1.** Cartilage size in *twist1a;twist1b* mutants.
DOI: https://doi.org/10.7554/eLife.37024.008
**Source data 2.** Severity of cartilage defects in combinatorial Twist1 and Tcf12 mutants.
DOI: https://doi.org/10.7554/eLife.37024.009
**Source data 3.** Adult viability of combinatorial Twist1 and Tcf12 mutants.
DOI: https://doi.org/10.7554/eLife.37024.010

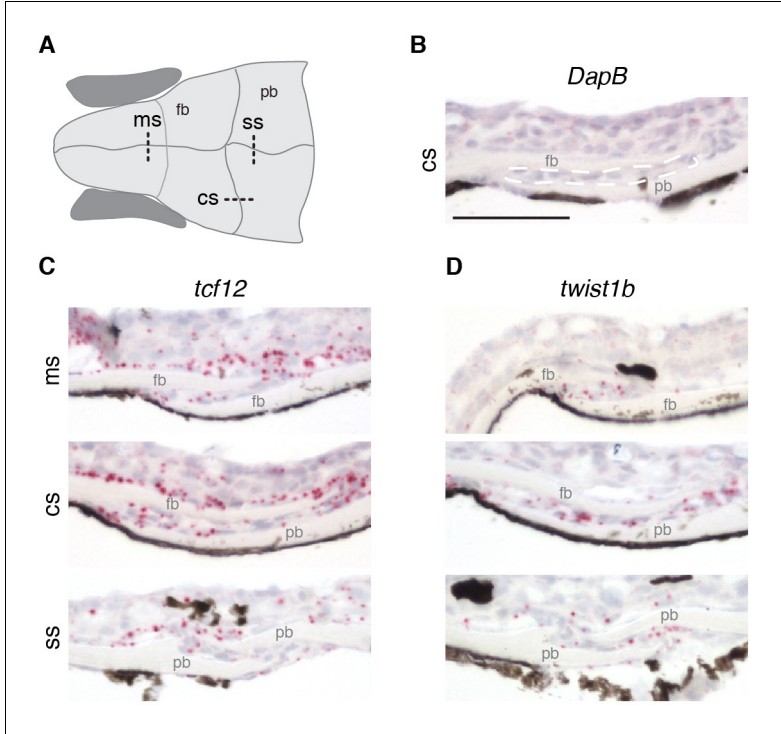

**Figure 3.** Expression of *tcf12* and *twist1b* in multiple sutures of zebrafish. (**A**) Schematic of the zebrafish skull depicting positions of sections (dotted lines) used for RNAscope in situ hybridizations. cs, coronal suture; ms, metopic suture; ss, sagittal suture; fb, frontal bone; pb, parietal bone. (**B–D**) In situ hybridizations on sections taken from zebrafish at 14 mm standard length. Red puncta indicate positive expression. *DapB* (**B**) was included as a negative control, with suture mesenchyme outlined in a dashed white line for reference. Expression of *tcf12* (**C**) and *twist1b* (**D**) was detected in the metopic, coronal and sagittal suture mesenchyme, with *tcf12* also expressed broadly outside the sutures. *n* = 3 for each experiment. Scale bar, 50 μm.
DOI: https://doi.org/10.7554/eLife.37024.011

together at the forming coronal suture, particularly in the medial region most commonly affected in mutants (*Figure 4B*). Likewise, the lack of enhanced medial-lateral growth of the parietal bones correlates with the sagittal suture being unaffected in mutants.

Subsequent sequential staining with Alizarin Red unveiled a marked decrease in bone growth in mutants from 10.25 ± 0.25 to 14 ± 0.5 mm at the future coronal but not the metopic or sagittal sutures (*Figure 4D*, *Figure 4—source data 3*, *Figure 4—source data 4*, *Figure 4—source data 5*). We further took advantage of the variable penetrance of suture defects to correlate the degree of bone stalling with later suture loss in individual *tcf12; twist1b* mutants. Consistently, synostotic sides exhibited greater reductions in earlier bone growth compared to mutant sides that had patent sutures. Thus, the degree to which the growth of the mutant parietal and frontal bones slows in the coronal zone predicts later loss of this suture. Reciprocally, the absence of bone stalling at the future metopic and sagittal zones is consistent with these sutures being unaffected in mutants.

## Altered dynamics of bone front cells in *tcf12; twist1b* mutant fish

We next examined the cellular mechanisms underlying altered bone growth in mutants. Analysis of an *sp7:EGFP* transgenic line, which labels Sp7[+] osteoblasts, revealed accelerated frontal and parietal bone fronts along the diagonal axis in *tcf12; twist1b* mutants versus sibling controls at 10 mm (*Figure 5—figure supplement 1*), consistent with our analysis using mineralization dyes. We then used BrdU incorporation in combination with anti-Sp7 antibody staining to assess the numbers of proliferative cells at the growing fronts of the frontal and parietal bones at an earlier 9 mm stage. Analysis of Sp7[+] osteoblasts revealed acceleration of frontal and parietal bone growth, with the leading edges of the bones appearing uneven (*Figure 5A*; additional examples in *Figure 5—figure*

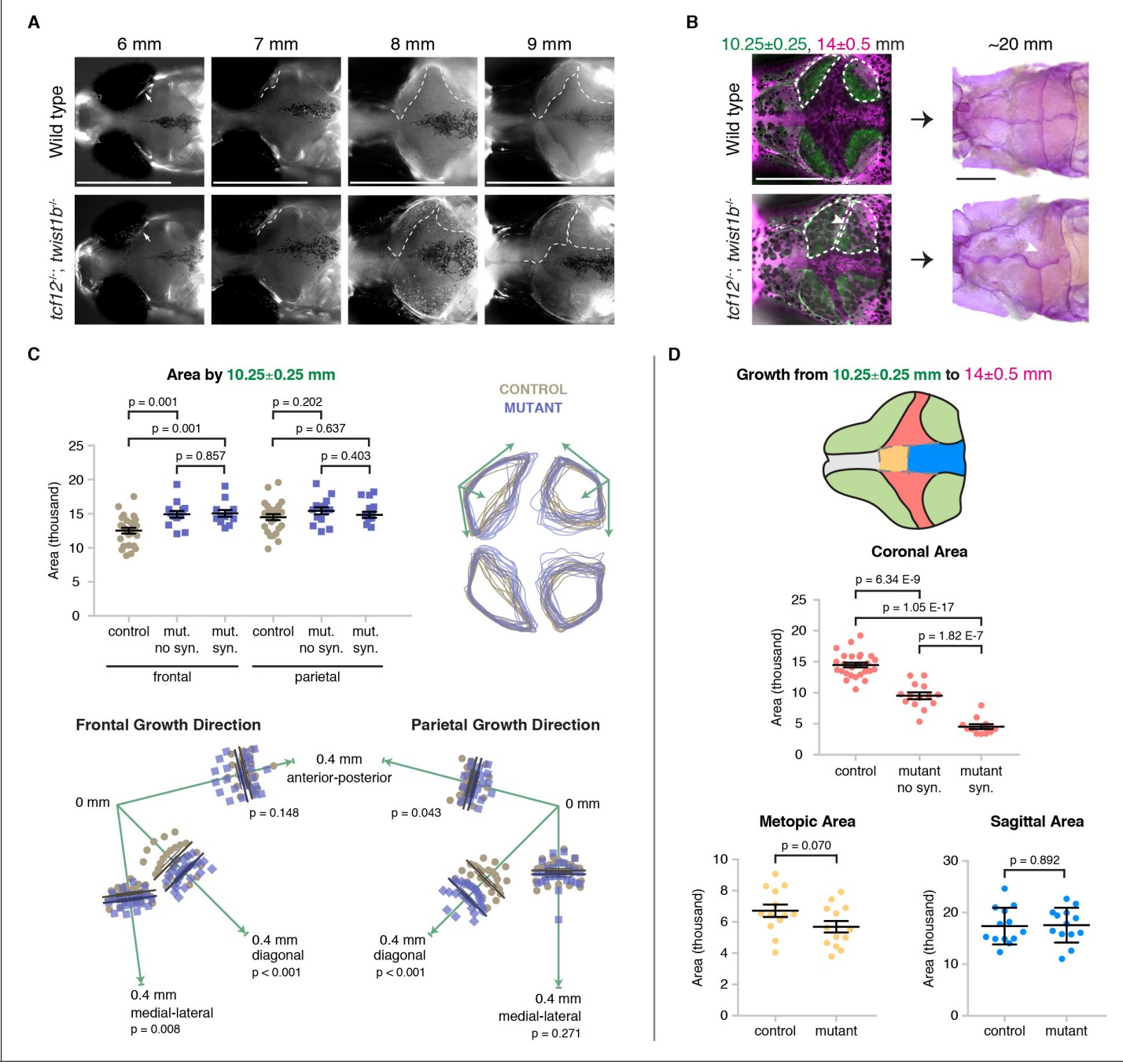

**Figure 4.** Altered bone growth dynamics precede craniosynostosis in mutant zebrafish. (**A**) Dorsal views of the developing skull bones in the same wild-type and mutant individuals across four developmental stages. Live fish were stained with Calcein Green at 6, 7, and 8 mm and Alizarin Red at 9 mm. For the right sides, arrows show initiation of the frontal bone at 6 mm and dashed lines show the frontal (left) and parietal (right) fronts at successive later stages. (**B**) Individual wild-type and *tcf12; twist1b* mutant fish were stained with Calcein Green at 10.25 ± 0.25 mm, recovered, and then stained again with Alizarin Red and imaged at 14 ± 0.5 mm. These same fish were then grown to 20 mm, at which stage they were fixed and stained again with Alizarin Red to assess suture patency. White dotted lines indicate bone generated by 10.25 mm. Arrowheads indicate missing coronal suture. Scale bars, 1 mm. (**C**) Quantification of calvarial bone growth. Bone produced by 10.25 ± 0.25 mm was calculated based on the area (µm) stained with Calcein Green (white outlines in B). At 10.25 mm, compared to control frontal bones (*n* = 12), *tcf12; twist1b* mutant frontal bones that developed synostosis later (*n* = 11) and those that did not (*n* = 11) showed similar increases in bone formation. Bone shape was assessed by overlaying tracings of posterior frontal bones and parietal bones for wild types and mutants. Specific growth directionality in the anterior-posterior, medial-lateral, and diagonal axes were measured and quantified (green arrows). (**D**) Bone growth from 10.25 ± 0.25 mm to 14 ± 0.5 mm was analyzed in respect to prospective suture zones. Growth in the metopic (yellow) and sagittal (blue) zones did not differ significantly in controls versus *tcf12; twist1b* mutants, which correlated with

*Figure 4 continued on next page*

*Figure 4 continued*

no defects in these sutures in mutants. In contrast, growth in the coronal zone was reduced in *tcf12; twist1b* mutants, with a more pronounced decrease in mutant sides that later developed synostosis. p values were determined by Student's t-tests; error bars represent standard error of the mean.

DOI: https://doi.org/10.7554/eLife.37024.012

The following source data is available for figure 4:

**Source data 1.** Quantification of skull bone area in mutants by 10.25 mm.
DOI: https://doi.org/10.7554/eLife.37024.013
**Source data 2.** Directionality of bone growth in mutants.
DOI: https://doi.org/10.7554/eLife.37024.014
**Source data 3.** Quantification of growth in coronal area.
DOI: https://doi.org/10.7554/eLife.37024.015
**Source data 4.** Quantification of growth in metopic area.
DOI: https://doi.org/10.7554/eLife.37024.016
**Source data 5.** Quantification of growth in sagittal area.
DOI: https://doi.org/10.7554/eLife.37024.017

*supplement 2*). After digitally extracting the bone fronts to avoid signals from the highly proliferative skin, we quantified the numbers of proliferative cells at the fronts. Along both the mutant frontal and parietal bone fronts, we observed an increase in the percentage of Sp7$^+$ osteoblasts undergoing proliferation, and a trend toward increased numbers of proliferative Sp7$^-$ cells just ahead of the bone fronts (*Figure 5B,C*, *Figure 5—source data 1*, *Figure 5—source data 2*). These results suggest that increased proliferation of early osteoblasts, and potentially also osteoprogenitors, contribute to the initial acceleration of bone growth across the mutant skull.

## Altered calvarial bone growth precedes coronal suture loss in *Tcf12$^{+/-}$; Twist1$^{+/-}$* mice

Given the similarity of coronal suture defects in *tcf12$^{-/-}$; twist1b$^{-/-}$* fish and *Tcf12$^{+/-}$; Twist1$^{+/-}$* mice, we investigated whether earlier alterations in calvarial bone growth might also prefigure suture loss in mice. At embryonic day (E) 13.5, when skull bone rudiments are first apparent, alkaline phosphatase staining revealed accelerated frontal and parietal bones that were in closer apposition in *Tcf12$^{+/-}$; Twist1$^{+/-}$* mutants versus wild-type sibling controls (*Figure 6A*). At birth, mutant frontal and parietal bones were more closely apposed than in controls and had abnormal shapes (*Figure 6B*). Next, we assessed proliferation rates and osteoblast density at the forming coronal suture (E14.5) and sagittal suture (E16.5) (*Figure 6C*, outlined regions of bottom panels). In mutants, we observed thicker bone fronts, as marked by Sp7$^+$ cells (*Figure 6—figure supplement 1*, *Figure 6-figure supplement 1-Source Data 1*), as well as a marked increase in the number of Sp7$^+$ osteoblasts at the bones fronts predestined for the coronal and sagittal sutures (*Figure 6D*, *Figure 6—source data 1*). We also observed an increase in the number of proliferative Sp7$^-$ cells immediately adjacent to osteoblasts at the bone fronts, with this increase more evident at the forming coronal versus sagittal suture (*Figure 6E*, *Figure 6—source data 2*). In contrast to fish, we noted a decrease in the number of proliferative osteoblasts at the forming coronal suture, and no change at the forming sagittal suture (*Figure 6F*, *Figure 6—source data 3*). These findings largely support a conserved role for Tcf12 and Twist1 in negatively regulating the number of proliferative cells at the growing bone fronts in fish and mice.

## Selective reduction of the osteoprogenitor pool at the mutant coronal suture

We next investigated whether the lack of continued bone growth at the mutant coronal fronts might reflect an exhaustion of osteoprogenitor cells. In mouse, sutural stem cells express *Prrx1* (*Wilk et al., 2017*) and *Gli1* (*Zhao et al., 2015*). *Grem1* also marks skeletal stem cells throughout the animal (*Worthley et al., 2015*), yet a role in the skull and sutures has not been previously examined. Using RNAscope in situ hybridization technology in zebrafish, we find that *prrx1a* is broadly expressed at the parietal and frontal bone fronts destined for the coronal and sagittal sutures, as well as the periosteum, at 10 mm, and in the sutures and periosteum at adult stages (*Figure 7A,B*). The expression

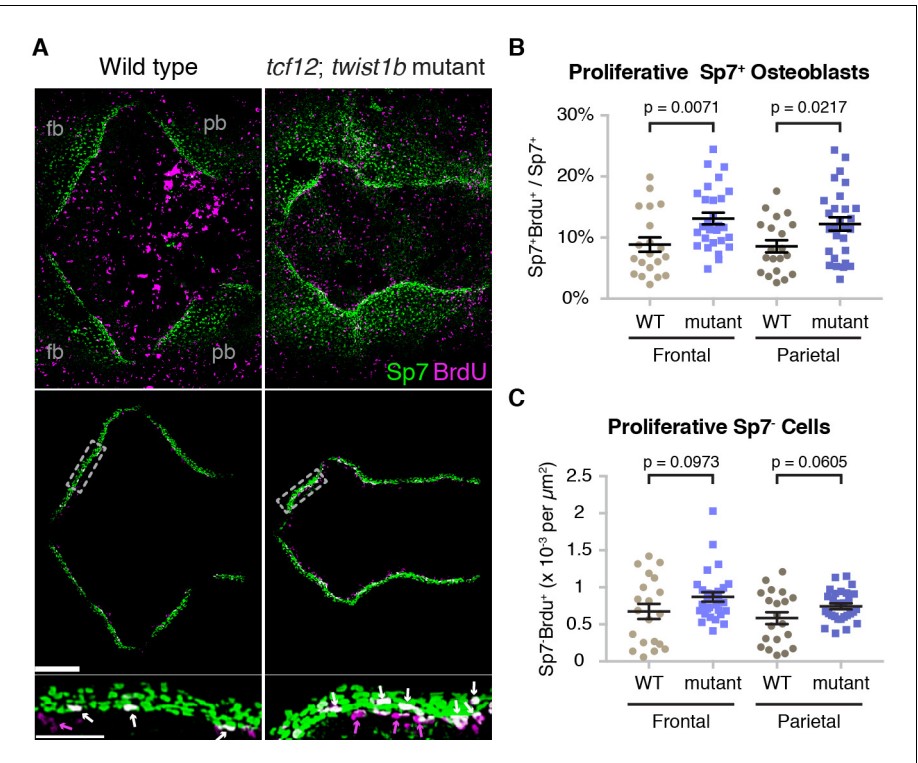

**Figure 5.** Altered proliferation and osteoblast production at mutant zebrafish bone fronts. (**A**) Dissected skullcaps were stained for BrdU (magenta) and Sp7 protein (green) at 9 mm. Top panels show maximum intensity projections of whole skull volumes, and middle panels are the same volumes but processed to extract the bone fronts (note that much of the BrdU staining in the center of the top images is in the skin). Bottom panels show enlarged regions of the osteogenic fronts (dotted rectangles). White arrows show proliferative osteoblasts (BrdU$^+$/Sp7$^+$) and magenta arrows show adjacent proliferative Sp7$^-$ cells. fb, frontal bone; pb, parietal bone. Scale bars, 300 μm for whole skull view, 100 μm for enlarged view. (**B, C**) Based on the extracted osteogenic fronts (middle panels in A), we quantified the percentage of Sp7$^+$ osteoblasts that were BrdU$^+$ (**B**) and the number of adjacent BrdU$^+$/Sp7$^-$ cells per area (**C**). Wild-type controls, n = 20; *tcf12; twist1b* mutants, n = 28. p values were determined by a Student's t-test; error bars represent standard error of the mean.

DOI: https://doi.org/10.7554/eLife.37024.018

The following source data and figure supplements are available for figure 5:

**Source data 1.** Quantification of proliferative sp7+ osteoblasts in mutant zebrafish.
DOI: https://doi.org/10.7554/eLife.37024.021

**Source data 2.** Quantification of proliferative sp7- cells in mutant zebrafish.
DOI: https://doi.org/10.7554/eLife.37024.022

**Figure supplement 1.** Accelerated bone fronts in *tcf12; twist1b* mutants transgenic for *sp7:EGFP*.
DOI: https://doi.org/10.7554/eLife.37024.019

**Figure supplement 2.** Additional examples of BrdU and Sp7 staining in *tcf12; twist1b* mutant zebrafish.
DOI: https://doi.org/10.7554/eLife.37024.020

of *gli1* and *grem1a* appears more restricted to the growing bone fronts and suture mesenchyme, although we also see more general periosteal expression at earlier stages. In *tcf12; twist1b* mutants, we still observe cells expressing *gli1*, *grem1a*, and *prrx1a* at the forming coronal and sagittal sutures, as well as within the periosetum (*Figure 7A,B*), with quantitation revealing similar levels of each gene on a per cell basis (*Figure 7—figure supplement 1A*, *Figure 7—figure supplement 1— source data 1*, *Figure 7—figure supplement 1—source data 2*). At the coronal suture, we observed that the mutant frontal and parietal bones were more closely apposed than in stage-matched wild-type siblings, which could possibly be a consequence of depleted progenitors at the bone fronts. In adult mutants with fused coronal sutures, we failed to detect *gli1$^+$*, *grem1a$^+$*, or *prrx1a$^+$* cells in the coronal suture region, whereas *prrx1a* was still expressed in the periosteum

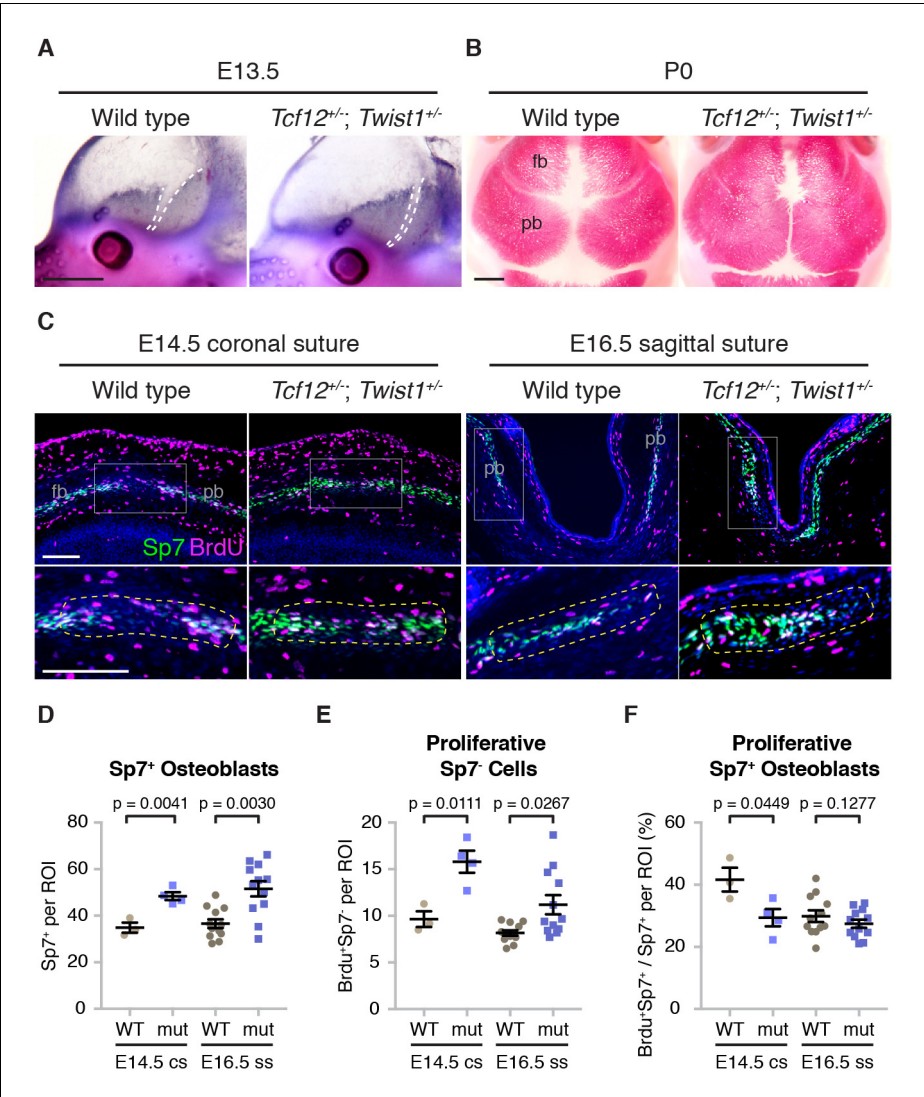

**Figure 6.** Altered bone growth dynamics in *Tcf12*+/-; *Twist1*+/- mice. (**A**) Lateral views of E13.5 mouse heads show Alkaline phosphatase staining of developing frontal and parietal bones. Dotted lines indicate the bone fronts that will form the coronal suture. Compared to wild types (*n* = 5), the fronts were accelerated in all *Tcf12*+/-; *Twist1*+/- mutants (*n* = 12). Scale bar, 1 mm. (**B**) Dorsal views of skull bones stained with Alizarin Red at birth (P0). Compared to wild types (*n* = 5), the fronts were closer together in all *Tcf12*+/-; *Twist1*+/- mutants (*n* = 11). Scale bar, 1 mm. (**C**) Sections of E14.5 coronal sutures and E16.5 sagittal sutures stained for BrdU (magenta), Sp7 protein (green), and DAPI (blue, nuclei). Boxed regions are magnified in lower panels, with yellow dotted lines indicating the regions of interest (ROI) used for quantification. fb, frontal bone; pb, parietal bone. Scale bar, 100 μm. (**D–F**) Quantification of Sp7+ osteoblasts per ROI (**D**), BrdU+Sp7- bone front cells per ROI (**E**), and the percentage of Sp7+ osteoblasts that are BrdU+ in the ROI (**F**). Cell counts were performed at the developing coronal sutures (cs, four wild types, three mutants) and sagittal sutures (ss, six wild types, six mutants). p values were determined by Student's t-tests; error bars represent standard error of the mean.

DOI: https://doi.org/10.7554/eLife.37024.023

The following source data and figure supplements are available for figure 6:

**Source data 1.** Quantification of Sp7+ osteoblast number in mutant mice.
DOI: https://doi.org/10.7554/eLife.37024.026
**Source data 2.** Quantification of proliferative Sp7- cells in mutant mice.
DOI: https://doi.org/10.7554/eLife.37024.027
**Source data 3.** Quantification of proliferative Sp7+ osteoblasts in mutant mice.
DOI: https://doi.org/10.7554/eLife.37024.028
**Figure supplement 1.** Increased calvarial bone front thickness in mutants.
*Figure 6 continued on next page*

*Figure 6 continued*

DOI: https://doi.org/10.7554/eLife.37024.024

**Figure supplement 1—source data 1.** Quantification of calvarial bone front thickness in mutant mice.

DOI: https://doi.org/10.7554/eLife.37024.025

(*Figure 7A*). Normal expression of all three markers was observed at the patent sagittal suture. We also examined Fgf signaling in zebrafish mutants, as the expression of *Fgfr2* has been reported to be altered in mouse *Twist1* heterozygous animals (*Rice et al., 2000*; *Connerney et al., 2006*). As with *gli1*, *grem1a*, and *prrx1a*, we still observed cells expressing *fgfr2* and the Fgf target gene *dusp6* at the mutant coronal suture (*Figure 7—figure supplement 1B*, *Figure 7—figure supplement 1—source data 3*), arguing against the calvarial phenotypes being due to wholesale loss of Fgfr2 signaling at the bone fronts.

Given the difficulty in quantitating the numbers of osteoprogenitors at the forming coronal suture zone of mutant zebrafish, owing in part to the small sizes of these bone fronts, we also examined putative progenitors in mutant mice. As in zebrafish, we observed cells expressing Gli1 and Grem1 protein at and around the fronts of the embryonic frontal and parietal bones. In $Tcf12^{+/-}; Twist1^{+/-}$ mice, we observed a marked reduction in the number of $Gli1^+$ and $Grem1^+$ putative progenitors at and around the developing coronal but not the sagittal bone fronts (*Figure 7C,D*). These findings highlight a conserved molecular signature of putative osteoprogenitors and sutural stem cells of zebrafish and mice and suggest, at least in mice, a selective exhaustion of osteoprogenitors at the developing coronal suture.

## Tissue-specific roles for Twist1 in calvarial bone growth and suture patency

To investigate whether Twist1 functions tissue-intrinsically for proper skull bone growth, we took advantage of the unique germ-layer origins of the mammalian frontal and parietal bones to remove one copy of *Twist1* in each tissue. At postnatal day (P) 21, we found that reduced dosage of *Twist1* in the neural-crest-derived precursors of the frontal bone, in *Wnt1-Cre;* Twist1$^{flox/+}$ mice, resulted in the overgrowth of the frontal bone relative to the parietal bone, which we quantified by measuring the ratio of the sagittal to metopic suture (*Figure 8*, *Figure 8—source data 1*). Reciprocally, removing one copy of *Twist1* from the mesoderm-derived parietal bone, in *Mesp1-Cre;* Twist1$^{flox/+}$ mice, resulted in its overgrowth relative to the frontal bone. Reduced dosage of *Twist1* in both the neural crest and mesoderm, in *Wnt1-Cre; Mesp1-Cre; Twist1*$^{flox/+}$ mice, normalized the relative sizes of the frontal and parietal bones and resulted in loss of the coronal suture, a phenotype not seen upon deletion of *Twist1* in neural crest or mesoderm alone. We conclude that Twist1 negatively regulates bone growth in both the neural-crest- and mesoderm-derived portions of the skull, and that *Twist1* must be mutated in not only the mesoderm-derived parietal bone and suture mesenchyme, but also the neural-crest-derived frontal bone, to impact coronal suture formation.

## Discussion

Discovery of a selective requirement for *tcf12* and *twist1b* in coronal suture formation in zebrafish has allowed us to gain a better understanding of the developmental basis of suture loss in Saethre-Chotzen syndrome. The similarity of coronal suture defects from humans to mice to zebrafish is striking, although each species displays unique dosage sensitivities to loss of *TWIST1* and *TCF*12. In humans, haploinsufficiency of *TWIST1* or *TCF12* can lead to suture loss. In mice, haploinsufficiency of *Twist1* also results in coronal suture loss, yet haploinsufficiency of *Tcf12* does not, despite enhancing the penetrance of suture defects in $Twist1^{+/-}$ mice (*Sharma et al., 2013*). In zebrafish, only loss of one of two Twist1 homologs (*twist1b*) along with loss of *tcf12* results in suture loss. It remains unclear why humans are more sensitive to Twist1 and Tcf12 dosage than mice and zebrafish, although a similar phenomenon has been observed with other synostosis genes (e.g. *JAG1*) (*Teng et al., 2017*).

Repeated live imaging of individual mutant fish revealed a strong correlation between the extent of altered bone growth and later suture loss. Whereas the initiation of the frontal and parietal bones was largely unaffected in mutants, we observed increased proliferation and osteoblast production at the mutant bone fronts in both fish and mice. There were some subtle differences between fish and

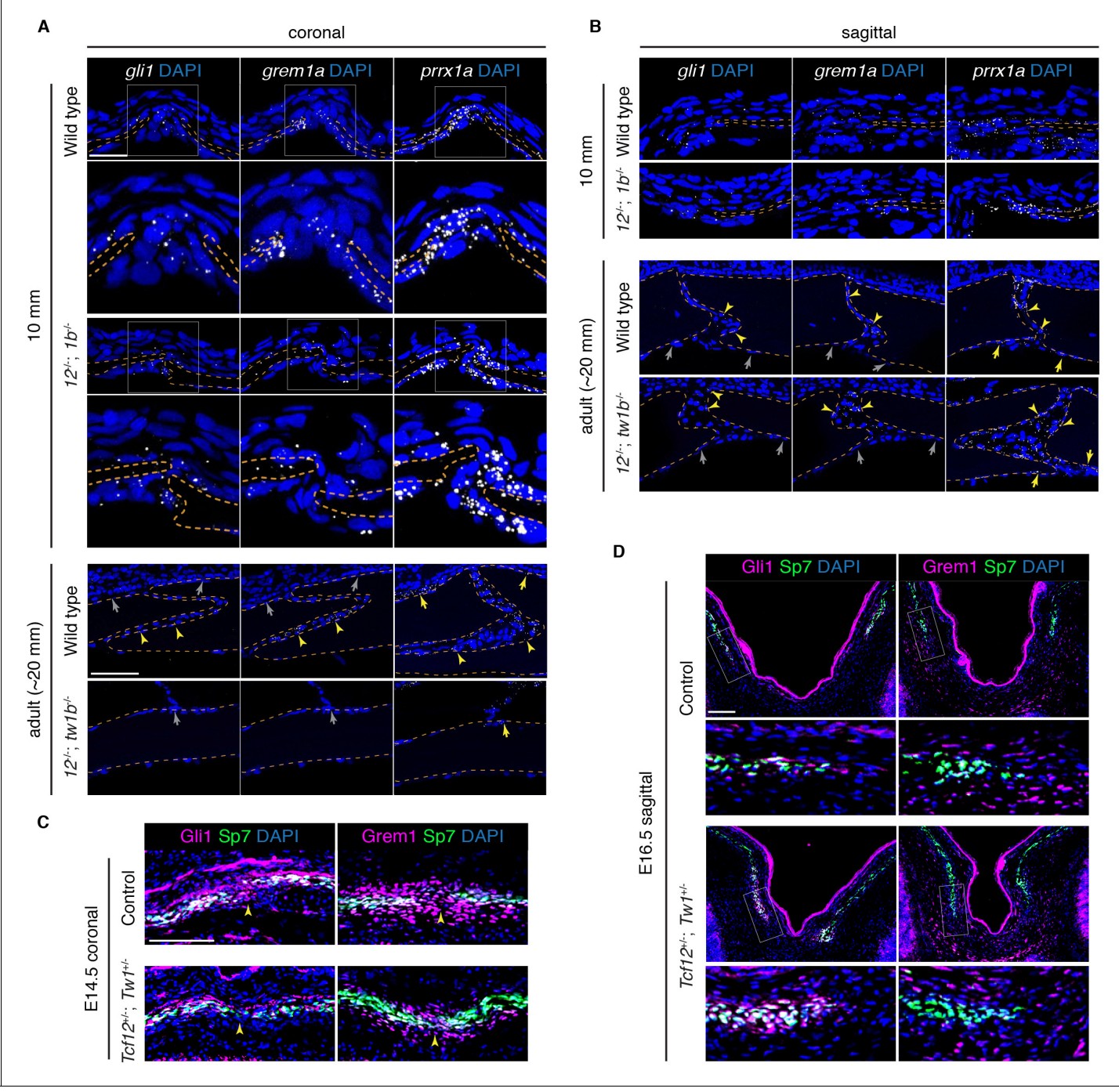

**Figure 7.** Reduced osteoprogenitor pool at the mutant coronal suture. (**A, B**) Sections of forming coronal and sagittal sutures of 10 mm fish and fully formed sutures of adult fish were assessed for *gli1*, *grem1a*, and *prrx1a* mRNA expression (white) by RNAscope in situ hybridization. Orange dotted lines indicate bones, and the boxed regions of the coronal suture regions are magnified below. For adult sutures, yellow arrowheads show expression in the suture mesenchyme, yellow arrows show expression of *prrx1a* in the periosteum, and grew arrows show lack of expression of *gli1* and *grem1a* in the periosteum. Scale bar at 10 mm stage, 20 μm; scale bar at adult stage, 50 μm. (**C, D**) Sections of E14.5 coronal sutures and E16.5 forming sagittal sutures of mice stained for Gli1/Grem1 (magenta) and Sp7 protein (green). Yellow arrowheads indicate progenitor regions in forming coronal sutures. Boxed regions of parietal bone fronts in the forming sagittal sutures are magnified in lower panels. Nuclei are stained blue by DAPI in all images. Scale bars, 100 μm.

DOI: https://doi.org/10.7554/eLife.37024.029

The following source data and figure supplements are available for figure 7:

**Figure supplement 1.** Quantification of progenitor marker and Fgf pathway transcripts in *tcf12*; *twist1b* mutants.

*Figure 7 continued on next page*

*Figure 7 continued*

DOI: https://doi.org/10.7554/eLife.37024.030

**Figure supplement 1—source data 1.** Quantification of *gli1,grem1a*, and *prrx1a* transcripts in coronal suture region of mutant fish.

DOI: https://doi.org/10.7554/eLife.37024.031

**Figure supplement 1—source data 2.** Quantification of *gli1,grem1a*, and *prrx1a* transcripts in sagittal suture region of mutant fish.

DOI: https://doi.org/10.7554/eLife.37024.032

**Figure supplement 1—source data 3.** Quantification of *fgfr2* and *dusp6* transcripts in coronal suture region of mutant fish.

DOI: https://doi.org/10.7554/eLife.37024.033

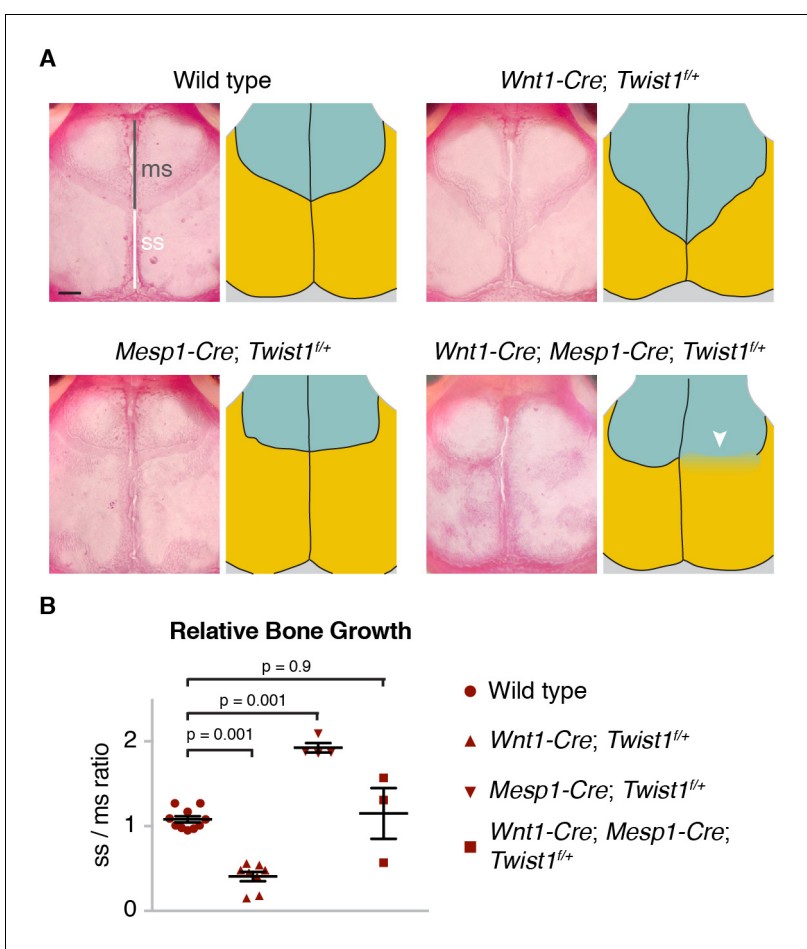

**Figure 8.** Tissue-autonomous bone overgrowth in *Twist1* conditional mutants. (**A**) Dorsal views of Alizarin-stained skulls of three-week-old mice. In the accompanying diagrams, turquoise indicates the neural-crest-derived frontal bones and gold the mesoderm-derived parietal bones. The relative lengths of the metopic suture (ms) and sagittal suture (ss) serve as a proxy for bone size. Compared to wild type (n = 0/10), *Wnt1-Cre; Twist1*$^{flox/+}$ (n = 0/8), and *Mesp1-Cre; Twist1*$^{flox/+}$ (n = 0/4); *Wnt1-Cre; Mesp1-Cre; Twist1*$^{flox/+}$ mice (n = 2/3) displayed coronal synostosis (arrowhead, average craniosynostosis index of 2.33). Scale bar, 1 mm. (**B**) Quantification of the relative length of the sagittal over the metopic suture. p values were determined by a one-way ANOVA with post-hoc Tukey-Kramer HSD test; error bars represent standard error of the mean.

DOI: https://doi.org/10.7554/eLife.37024.034

The following source data is available for figure 8:

**Source data 1.** Ratio of the sagittal to metopic suture length in mutant mice.

DOI: https://doi.org/10.7554/eLife.37024.035

mice, with larger increases in proliferative osteoblasts in mutant fish than mice, which might reflect species-specific or staging differences. Nonetheless, mutants in both species displayed accelerated growth of both the frontal and parietal bones, which often resulted in abnormal shapes likely due to growth variations along individual bone fronts. In particular, we found that increased diagonal growth in mutants brought the parietal and frontal bones together at the prospective medial regions of the coronal suture much earlier than in wild type animals, which correlates with the medial region of the coronal suture being most commonly fused in mutants. This altered directional growth might be one reason why the coronal suture is preferentially affected in both zebrafish and mouse Twist1/ Tcf12 mutants, despite the different origins of the coronal suture in these species.

Another prominent finding was a lack of continued bone growth at the future coronal but not other sutures in mutants, with the degree of bone stalling predicting synostosis in individuals. We identified *Gli1/gli1* and *Grem1/grem1a* as conserved markers of putative osteoprogenitors in both the growing bone fronts and mature sutures of mice and zebrafish, although both genes display broader expression at early stages of calvarial bone development, including in some periosteal cells. Future lineage tracing experiments should help reveal the extent to which *gli1* in fish and *Grem1/ grem1a* in both species mark similar populations of embryonic osteoprogenitors and postnatal sutural stem cells. A previous study had observed reduced expression of *Gli1* in the sutures of adult *Twist1$^{+/-}$* mice (*Zhao et al., 2015*). Here, we extend this finding to embryonic stages when the coronal suture is forming, and uncover the existence of Grem1+ cells in and around the bone fronts of the nascent coronal suture that become depleted in mutant mice. Whether similar osteoprogenitors become exhausted at the developing coronal suture of zebrafish remains to be determined, as it was difficult to precisely quantify the numbers of these progenitors by RNA expression alone. One possibility is that reduced Twist1 and Tcf12 function alters the balance between long-term sutural stem cells (i.e. those marked by *Gli1*) and proliferative osteoblasts and their immediate progenitors. In such a model, the early increase in osteoblast production would come at the expense of long-term progenitors, thus leading to a later failure of continued bone growth and a loss of the sutural stem cells that would normally separate the skull bones. In addition to further verifying this model, an important next step will be to determine why the fronts of the parietal and frontal bones abutting the future coronal suture are most sensitive to progenitor exhaustion in mutants. We did not observe preferential expression of *tcf12* and *twist1b* at the coronal suture in fish, consistent with similarly broad sutural expression of *Twist1* in mice (*Rice et al., 2000*). Instead, osteoprogenitors in the coronal zone could be fewer in number at initial stages and/or more sensitive to loss of Tcf12 and Twist1, for example due to compensation by related genes at other sutures.

Further support for suture defects arising from much earlier changes in bone growth come from our conditional *Twist1* deletion experiments in mouse. Whereas the frontal bone arises from neural crest and the parietal bone from mesoderm, the mesenchyme within the postnatal coronal suture is largely mesoderm-derived (*Yen et al., 2010*). However, suture loss was only observed upon conditional deletion of one allele of *Twist1* from both the embryonic mesoderm and neural crest. This finding is inconsistent with Twist1 functioning solely in the mesoderm-derived postnatal suture mesenchyme for suture patency, instead suggesting that misregulated growth of both the frontal and parietal bones is required to later disrupt the coronal suture in *Twist1* mutants.

Our study highlights a selective role for Tcf12 in the later growth of the skull bones and patency of the coronal suture. In contrast to animals lacking Twist1 homologs, zebrafish lacking *tcf12* do not display embryonic lethality or defects in ectomesenchyme and facial cartilage formation. Instead, loss of *tcf12* partially suppresses the facial cartilage defects and lethality of *twist1a; twist1b* mutants. Suppression of embryonic defects could be due to loss of competition of Tcf12 with other Twist binding partners, such as Hand2 (*Firulli et al., 2005*). In this scenario, maternal Twist1a/b and/or other Twist family members (e.g. *twist2* and *twist3*) could compensate for lack of zygotic Twist1a/b. Loss of *tcf12* would then allow remaining Twist proteins to more effectively form homodimers or alternate heterodimers important for embryogenesis. Whereas a previous report indicates that Tcf12 and Twist1 can form heterodimers (*Connerney et al., 2006*), it is also possible that Tcf12 has Twist1-independent functions that antagonize Twist1 during embryogenesis. Future efforts to directly visualize Tcf12-Twist1 complexes will help to resolve how Tcf12 promotes Twist1 function during skull bone growth while counteracting it during embryonic neural crest development.

Development of homologous structures often employs ancestrally conserved gene regulatory networks. For example, a requirement for Pax6 genes in eye development in a wide range of animals

has been used to argue for deep homology of eye structures (*Gehring and Ikeo, 1999*). Here, we show remarkably specific loss of a single anatomical structure, the coronal suture, in zebrafish and mice lacking Tcf12 and Twist1, despite this suture occurring at a mesoderm/mesoderm interface in zebrafish and a neural-crest/mesoderm interface in mice. Hence, sensitivity of the coronal suture in Saethre-Chotzen syndrome is unlikely to be due to its location at a unique neural-crest/mesoderm interface. In addition, there are several other sutures occurring at a neural-crest/mesoderm interface in mice and fish that are not affected by loss of Tcf12 and Twist1. There has been on-going debate, given these distinct tissue boundaries, as to whether coronal sutures are truly homologous across vertebrates (*Maddin et al., 2016*). Our data indicate that the conserved genetic dependence of the coronal suture in fish and mammals likely reflects a similar sensitivity to early bone growth changes, perhaps owing to similar developmental and anatomical constraints irrespective of the embryonic origins of the bones flanking this suture.

## Materials and methods

**Key resources table**

| Reagent type (species) or resource | Designation | Source or reference | Identifiers | Additional information |
|---|---|---|---|---|
| Genetic reagent (*D. rerio*) | sp7:EGFP | PMID: 20506187 | RRID: ZFIN_ZDB-GENO-100402-2 | Zebrafish International Resource Center |
| genetic reagent (*D. rerio*) | tcf12−/− | this paper | | allele el548; see *Supplementary file 2* |
| Genetic reagent (*D. rerio*) | twist1a−/− | this paper | | allele el571; see *Supplementary file 2* |
| Genetic reagent (*D. rerio*) | twist1b−/− | this paper | | allele el570; see *Supplementary file 2* |
| Genetic reagent (*M. musculus*) | Tcf12−/− | PMID: 23354436 | | Dr. Robert Maxson (University of Southern California) |
| Genetic reagent (*M. musculus*) | Twist1−/− | PMID: 7729687 | RRID:IMSR_JAX:002221 | Dr. Richard Behringer (University of Texas, M. D. Anderson Cancer Center) |
| Genetic reagent (*M. musculus*) | Twist1flox | PMID: 19414008 | RRID:MMRRC_016842-UNC | Dr. Patrick Tam (Children's Medical Research Institute, The University of Sydney) |
| Genetic reagent (*M. musculus*) | Wnt1-Cre | PMID: 9843687 | RRID:IMSR_JAX:003829 | Dr. Henry Sucov (University of Southern California) |
| Genetic reagent (*M. musculus*) | Mesp1-Cre | PMID: 10393122 | | Dr. Sachicko Iseki (Tokyo Medical and Dental University) |
| Antibody | rabbit anti-Osx/Sp7 | Santa Cruz Biotechnology | cat.#: sc-22536-r; RRID: AB_831618 | (1:300) |
| Antibody | rat anti-BrdU | Bio-Rad Laboratories | cat.#: MCA2060 GA; RRID: AB_10545551 | (1:100–150) |
| Antibody | goat anti-Grem1 | Thermo Fisher Scientific | cat.#: PA5-47973; RRID: AB_2610125 | (1:40) |
| Antibody | goat anti-Gli1 | R and D Systems | cat.#: AF3455; RRID: AB_2247710 | (1:40) |
| Antibody | goat anti-rat FITC | Santa Cruz Biotechnology | cat.#: sc-2011; RRID: AB_631753 | (1:200) |
| Antibody | goat anti-rabbit Alexa Fluor 568 | Thermo Fisher Scientific | cat.#: A-11011; RRID: AB_143157 | (1:200–500) |
| Antibody | donkey anti-goat Alexa Fluor 488 | Abcam | cat.#: ab150129; RRID: AB_2687506 | (1:200) |

*Continued on next page*

*Continued*

| Reagent type (species) or resource | Designation | Source or reference | Identifiers | Additional information |
|---|---|---|---|---|
| Antibody | goat anti-rat Alexa Fluor 633 | Thermo Fisher Scientific | cat.#: A21094; RRID: AB_2535749 | (1:500) |
| Sequence-based reagent | RNAscope Probe - Dr-tcf12-C2 | Advanced Cell Diagnostics | cat.#: 517031-C2 | |
| Sequence-based reagent | RNAscope Probe - Dr-twist1b | Advanced Cell Diagnostics | cat.#: 413121 | |
| Sequence-based reagent | RNAscope Probe - Dr-gli1-C3 | Advanced Cell Diagnostics | (not yet in catalog) | |
| Sequence-based reagent | RNAscope Probe - Dr-grem1a | Advanced Cell Diagnostics | cat.#: 535291 | |
| Sequence-based reagent | RNAscope Probe - Dr-prrx1a | Advanced Cell Diagnostics | cat.#: 535321 | |
| Sequence-based reagent | RNAscope Probe - Dr-fgfr2 | Advanced Cell Diagnostics | cat.#: 420961 | |
| Sequence-based reagent | RNAscope Probe - Dr-dusp6-C3 | Advanced Cell Diagnostics | cat.#: 515021-C3 | |
| Commercial assay or kit | RNAscope 2.5 HD Assay – RED | Advanced Cell Diagnostics | cat.#: 322350 | |
| Commercial assay or kit | RNAscope Multiplex Fluorescent Kit v2 | Advanced Cell Diagnostics | cat.#: 323110 | |
| Chemical compound, drug | Alizarin Red S | Amresco | cat.#: 9436–25G | live staining: 1 mg / 30 mL |
| Chemical compound, drug | Calcein | Thermo Fisher Scientific | cat.#: C481 | live staining: 1 mg / 10 mL |
| Chemical compound, drug | Alcian Blue | Anatech LTD | cat.#: 862 | |

## Experimental model and subject details

### Animals

The University of Southern California Institutional Animal Care and Use Committee approved all animal experiments, and all methods were performed in accordance with the relevant guidelines and regulations. Zebrafish (*Danio rerio*) embryos were raised in Embryo Medium (**Westerfield, 2007**) at 28.5°C. Juvenile and adult fish were housed in groups of 10–15. Mutant lines were maintained on a mixed Tubingen wild-type (**Haffter et al., 1996**), *casper* (**White et al., 2008**), and *sp7:EGFP* background. Three targeted mutant lines (*tcf12$^{el548}$*, *twist1a$^{el571}$*, and *twist1b$^{el570}$*) were generated for this study (see below). Lines were propagated by genotyping fin biopsies. As we found that *twist1a* loss did not effect the penetrance or expressivity of suture loss in *tcf12$^{-/-}$*; *twist1b$^{-/-}$* fish (**Supplementary file 1**), we pooled *tcf12$^{-/-}$*; *twist1b$^{-/-}$* fish with any *twist1a* genotype for the experiments. Mice (*Mus musculus*) were housed in cages with no more than five adults or three adults with one litter per cage. The *Tcf12* (**Wojciechowski et al., 2007**), *Twist1* (**Chen and Behringer, 1995**), conditional *Twist1-flox* (**Bildsoe et al., 2009**), *Wnt1-cre* (**Danielian et al., 1998**), and *Mesp1-cre* (**Saga et al., 1999**) alleles were genotyped as described.

## Method details

### Generation of zebrafish mutant lines

Zebrafish mutant for *tcf12*, *twist1a*, or *twist1b* were generated with TALEN-based targeted mutagenesis. TALEN constructs were generated using the PCR-based platform (**Sanjana et al., 2012**) and digested with StuI (New England Biolabs, Ipswich, MA). RNAs were synthesized from linearized constructs using the mMessage mMachine T7 Ultra kit (Ambion/Life Technologies, Carlsbad, CA, USA). TALEN RNAs were injected at 100 ng/μl into 1-cell-stage embryos, and we identified mosaic germline founders by sequencing their progeny. The *tcf12$^{el548}$* allele includes base pair changes and insertions that disrupt a common exon shared in all protein-coding transcript variants. The *twist1a$^{el571}$*

and *twist1b*[el570] alleles are deletions that interrupt the coding sequence close to the translation start site. All three alleles are frame-shift mutations that result in premature stop codons upstream of the helix-loop-helix DNA-binding domains. Detailed sequences of TALEN targets and genotyping primers are listed in *Supplementary file 2*.

## Skull preparations

Adult zebrafish were fixed overnight at 4°C with 4% paraformaldehyde, washed with 0.5% KOH for at least half an hour, cleared with 3% $H_2O_2$ in 0.5% KOH for several hours until pigmentation was removed, washed with 35% $NaBO_4$ for at least half an hour, incubated with 1% trypsin in 35% $NaBO_4$ for several hours until tissue was reasonably cleared, washed with 10% glycerol in 0.5% KOH for at least one hour, stained with 0.02% Alizarin Red S (Amresco 9436) pH 7.5 in 10% glycerol and 0.5% KOH overnight, washed with 50% glycerol/0.5% KOH until residual stain was removed, and stored in 100% glycerol. Skullcaps were dissected and imaged using a Leica S8 APO stereomicroscope.

The heads of newborn mice were skinned and cleared with 1% KOH for 1 to 2 days, stained with 2% Alizarin Red S in 1% KOH until mineralized bone was red, and stored in 100% glycerol. Mouse skulls were imaged using a Leica MZ125 stereomicroscope.

## Micro-computed tomography

Data was collected on a Nikon Metrology Xt S 225 ST with the following parameters: energy at 120kV; current at 26 uA; 3141 projections at two frames/sec and averaging two frames; no filter; and at 6 µm resolution. Two-dimensional slices were rendered into three-dimensional reconstructions using Arivis (Phoenix, AZ).

## Paraffin embedding, sectioning, and tissue histology

Whole zebrafish were embedded into paraffin according to standard protocol. Briefly, fish heads were fixed with 4% paraformaldehyde at 4°C or 10% neutral buffered formalin at room temperature overnight, washed with PBS for half an hour each time, and then separated into heads and trunks. Heads were decalcified with 20% EDTA pH 8.0 for 10 days at room temperature, washed with DEPC-treated water, dehydrated through a series of sequentially increasing ethanol to DEPC water ratios, treated through a series of sequentially increasing Hemo-De (xylene substitute) to ethanol ratios, washed in 50% paraffin in Hemo-De at 65°C for one hour, incubated at 65°C in 100% paraffin overnight, embedded in paraffin in molds, and allowed to solidify at room temperature. Paraffin blocks were cut into 5 µm sections using a Shandon Finesse Me + microtome (cat no. 77500102) and collected on superfrost plus slides (Thermo Fisher Scientific).

For tissue histology, hematoxylin and eosin staining was performed according to standard protocol. Briefly, sections were deparaffinized in xylene and ethanol, rinsed in water, stained with hematoxylin, rinsed in 4% glacial acetic acid, rinsed with water, washed with blueing solution, rinsed with water, dried with ethanol, stained with eosin, washed with ethanol and Hemo-De, and mounted with cytoseal.

For whole-mount alkaline phosphatase staining, E13.5 mouse heads were fixed in 4% paraformaldehyde in PBS, bisected mid-sagitally after fixation, and stained with NBT and BCIP. Sections were imaged on a Leica DM2500 compound microscope.

## In situ hybridization

Colorimetric *in situ* hybridization using a *sox10* digoxigenin-labeled riboprobe was performed on 20 hpf zebrafish embryos as described (*Cox et al., 2012*). Briefly, embryos were fixed, dehydrated with an increasing methanol series, and stored in 100% methanol at −20°C until use. Embryos were then rehydrated with a decreasing methanol series, treated with 1 µg/ml Proteinase K for 6.6 min, post-fixed with 4% PFA, incubated with hybridization buffer (50% formamide, 5X SSC, 100 µg/ml yeast RNA, 50 µg/ml heparin, 0.125% Tween-20, citric acid to pH 6), hybridized with probe, washed with solution series with decreasing formamide and SSC, incubated in blocking solution (5 mg/ml BSA, 5% sheep serum), treated with sheep digoxigenin-AP at 1:10,000, stained with NBT and BCIP (Roche), and color was allowed to develop for two and a half hours. Stained embryos were imaged on a Leica DM2500 compound microscope. RNAscope in situ hybridization was performed with the

RNAscope 2.5 HD Assay – RED (Advanced Cell Diagnostics, Newark, CA) or the RNAscope Multiplex Fluorescent Kit v2 (Advanced Cell Diagnostics, Newark, CA) according to the manufacturer's protocol for formalin-fixed paraffin-embedded sections. Probes include *twist1b* (C1), *tcf12* (C2), *prrx1a* (C1), *grem1a* (C1), *gli1* (C3), *fgfr2* (C1), and *dusp6* (C3).

## Zebrafish larvae skeletal staining

Alcian Blue (cartilage) and Alizarin Red S (bone) staining was performed on 5 dpf zebrafish larvae as previously described (*Walker and Kimmel, 2007*). Briefly, larvae were fixed for one hour in 2% paraformaldehyde, rinsed in 100 mM Tris pH 7.5 in 10 mM $MgCl_2$, incubated overnight in Alcian Blue solution (0.04% Alcian Blue, 80% ethanol, 100 mM Tris pH 7.5, 10 mM $MgCl_2$), rehydrated through a sequentially decreasing series of ethanol to 100 mM Tris pH 7.5 and 10 mM $MgCl_2$ ratio, bleached with 3% $H_2O_2$ in 0.5% KOH under a lamp, washed with 25% glycerol in 0.1% KOH, stained with Alizarin Red S (0.01% Alizarin Red, 25% glycerol, 100 mM Tris pH 7.5), and de-stained with 50% glycerol in 0.1% KOH. Dissected jaw cartilages were mounted in 50% glycerol on a slide and imaged with a Leica DM2500 compound microscope.

## Sequential live staining and imaging

Fish were anesthetized, measured for body length, and recovered in Calcein Green stain (3 mg/30 ml, Molecular Probes C481) overnight in the dark. The following day, fish were washed in fish system water at least twice for one hour each time before imaging with a Zeiss AxioZoom microscope and returned to tanks on system. This process was repeated at a later time point with Alizarin Red S (1 mg/30 ml).

## BrdU treatments and immunohistochemistry

Anesthetized fish were measured for body length and incubated in 4.5 mg/ml BrdU solution (B5002, Sigma Aldrich) for two hours in the dark. Fish were then transferred to fish system water or embryo medium for 15 min before euthanasia. Heads were fixed in 4% paraformaldehyde overnight before skullcaps were dissected and stored in PBS at 4°C. For mouse embryos at E14.5 and E16.5, BrdU was injected into the pregnant female (200 µg/g body weight) 2 hr prior to dissection. Heads of embryos were embedded in OCT medium (Histoprep, Fisher Scientific) before sectioning. Frozen sections were cut at 10 µm. Immunohistochemistry was performed using rat anti-BrdU (MCA2060 GA, Bio-Rad), rabbit anti-Osx/Sp7 (sc-22536-r, Santa Cruz), goat anti-Grem1 (PA5-47973, Thermo Fisher), or goat anti-Gli1 (AF3455, R and D Systems) diluted in 1% BSA/PBS and incubated overnight at 4°C. Detection of primary antibodies was performed by incubating goat anti-rat FITC (sc-2011, Santa Cruz), goat anti-rabbit Alexa Fluor 568 (A-11011, Thermo Fisher Scientific), or donkey anti-goat Alexa Fluor 488 (ab150129, Abcam) for 1 hr at room temperature followed by DAPI counterstaining.

## Quantitation and statistical analyses

Severity of coronal synostosis was quantified using a coronal synostosis scoring index that was adapted from Oram and Gridley's craniosynostosis index (*Oram and Gridley, 2005*). Left and right sutures were given scores of 0 (no fusion), 1 (<50% fused), 2 (≥50% fused), or 3 (completely fused). A composite score for each animal was calculated by adding the sum of both suture scores. The index value was determined by the sum of composite scores in each genotype group divided by the number of animals in the group.

The degree of jaw cartilage defect was determined by the number of cartilage elements affected, with more affected equating to a higher grade of severity. Flat-mounted cartilages were measured for area using Fiji.

Directional bone growth was quantified using Fiji. For bone produced by 10.25 mm and subsequent growth by 14 mm, bone regions were drawn freehand and measured for area. For assessment of BrdU[+] and Sp7[+] cells, tiled z-stacks were captured of stained zebrafish skulls caps using a Zeiss LSM 800. The bone fronts were then digitally extracted using a 30-pt. brush in Amira software, and cells were manually counted in a 3D view on Imaris software. In mouse coronal suture sections, the region of interest was determined by a defined length across the bone fronts and suture. In mouse sagittal suture sections, defined lengths included several cell diameters medial from the last Sp7[+]

cell at the bone fronts and accounted for bone curvature. Five sections per animal were quantified and averaged. For bone thickness quantifications, a perpendicular measurement across the broadest point of Sp7+ cells was measured for each section. All measurements and counting were completed in Fiji. For the quantification of RNAscope experiments, regions of interests were defined by the edge of the calvarial bones (imaged by DIC microscopy) in Fiji. The coronal progenitor region was defined as the space between the neighboring frontal and parietal bones, and the sagittal progenitor region was defined as an approximately three-cell layer length region ahead of the parietal bone. RNAscope signal was quantified across a Z-stack projection using the 3D Object Counter tool, and measurements above a pre-defined unit of one transcript were adjusted to account for closely packed transcripts. The final transcript count was normalized to the area of the region of interest. Three sections per animal were averaged for each probe. Unpaired t-tests were performed for all statistical analyses, and all samples were scored blindly.

## Acknowledgements

We thank Megan Matsutani and Jennifer DeKoeyer Crump for fish care; Seth Ruffins, Director of the Microscopy Core Facility of USC Stem Cell, for assistance in rendering micro-CT reconstructions; and members of the Maxson, Crump, and Yang Chai labs for insightful discussions. This work was supported by NIH grant R01DE026339 to REM and JGC, NIH grant R35DE027550 to JGC, NIH training fellowships F31DE024031 and T90DE021982 to CST, and the Helen Hay Whitney Postdoctoral fellowship to DTF.

## Additional information

### Funding

| Funder | Grant reference number | Author |
| --- | --- | --- |
| National Institute of Dental and Craniofacial Research | F31DE024031 | Camilla S Teng |
| National Institute of Dental and Craniofacial Research | T90DE021982 | Camilla S Teng |
| Helen Hay Whitney Foundation | | D'Juan T Farmer |
| National Institute of Dental and Craniofacial Research | R01DE026339 | Robert E Maxson Jr. J Gage Crump |
| National Institute of Dental and Craniofacial Research | R35DE027550 | J Gage Crump |

The funders had no role in study design, data collection and interpretation, or the decision to submit the work for publication.

### Author contributions

Camilla S Teng, Conceptualization, Investigation, Methodology, Writing—original draft, Writing—review and editing; Man-chun Ting, Formal analysis, Investigation; D'Juan T Farmer, Conceptualization, Data curation, Formal analysis, Funding acquisition, Investigation, Methodology, Writing—review and editing; Mia Brockop, Data curation, Formal analysis, Investigation, Visualization, Methodology; Robert E Maxson, Conceptualization, Formal analysis, Supervision, Funding acquisition, Writing—original draft, Writing—review and editing; J Gage Crump, Conceptualization, Formal analysis, Supervision, Funding acquisition, Writing—original draft, Project administration, Writing—review and editing

### Author ORCIDs

J Gage Crump http://orcid.org/0000-0002-3209-0026

### Ethics

Animal experimentation: This study was performed in strict accordance with the recommendations in the Guide for the Care and Use of Laboratory Animals of the National Institutes of Health. The

University of Southern California Institutional Animal Care and Use Committee (IACUC) approved all animal experiments (protocol #20552), and all methods were performed in accordance with the relevant guidelines and regulations.

## Decision letter and Author response

Decision letter https://doi.org/10.7554/eLife.37024.040
Author response https://doi.org/10.7554/eLife.37024.041

# Additional files

### Supplementary files

• Supplementary file 1. Summary of phenotypes observed in combinatorial zebrafish mutants
DOI: https://doi.org/10.7554/eLife.37024.036

• Supplementary file 2. TALEN targeting and mutant genotyping
DOI: https://doi.org/10.7554/eLife.37024.037

• Transparent reporting form
DOI: https://doi.org/10.7554/eLife.37024.038

### Data availability

All data generated or analysed during this study are included in the manuscript and supporting files.

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
