## [Decision Letter]

Thank you for sending your article entitled "Altered progenitor dynamics prefigure craniosynostosis in a zebrafish model of Saethre-Chotzen syndrome" for peer review at *eLife*. Your article is being evaluated by three peer reviewers, and the evaluation has been overseen by a Reviewing Editor and Marianne Bronner as the Senior Editor. The reviewers have opted to remain anonymous.

We have included the detailed reviews below. All three of the reviewers commend the basic finding that Twist1/Tcf12 is involved in synostosis, and that the zebrafish is therefore a useful model for studying this syndrome. However, at the same time there is a major concern about a lack of a mechanistic explanation for why Tcf12 leads to the defects you have observed. After discussing this with each of the reviewers, our general feeling is that for us to consider proceeding at *eLife*, there would need to be a more compelling mechanistic explanation for the observed effects.

Based on these serious concerns, I would like to inquire as to whether you have additional mechanistic data that you think would address this.

*Reviewer #1:*

This manuscript from Teng et al. describes the effect of loss of Twist1 and Tcf12 on cranial bone growth and progenitor proliferation in the zebrafish and mouse models. It has been known for some time that heterozygous mutations in *Twist1* result in coronal synostosis in Saethre-Chotzen syndrome in humans and mice, and more recently, has been demonstrated that Tcf12 is a Twist1 binding partner that is also involved in coronal synostosis in humans and mice. Here, the authors demonstrate that as in mice, Twist1 and Tcf12 interact in zebrafish such that *tcf12^-/-^; twist1b^-/-^* exhibit coronal synostosis. Using successive bone staining in zebrafish the authors characterize bone growth in these mutants and find that though bone growth is slightly elevated in the earliest phase of formation of the development of the calvarial bones, bone growth is diminished later; these changes correlate with the incidence of synostosis. The number of proliferative osteoprogenitors and osteoblasts is increased in zebrafish, but in mouse only osteoprogenitors have an increased proliferation, though the total number of osteoblasts increases. Finally the authors demonstrate that the craniosynostosis phenotype requires loss of Twist1 from both the parietal and frontal bone. Together, these careful analyses make a compelling case that Twist1 and Tcf12 control bone progenitor cell dynamics to regulate normal growth of the calvaria and prevent craniosynostosis. The zebrafish work suggests that there is nothing unique about the neural crest/mesoderm boundary, but rather suggests that there may be directionality in the way that Twist1/Tcf12 regulate bone growth, though this possibility is not explored and the question of why the coronal suture is specifically synostosed remains uncertain. Nevertheless, these findings advance our understanding of the developmental function of Twist1 and Tcf12 in craniosynostosis in zebrafish and mice and will be of broad interest to the craniofacial development and disease communities. Additionally, these studies validate the zebrafish as a powerful model system for the study of the developmental mechanisms underlying craniosynostosis.

*Reviewer #2:*

The manuscript 'Altered progenitor dynamics prefigure craniosynostosis in a zebrafish model of Saethre-Chotzen syndrome' by Teng et al. presents interesting new findings concerning animal models of coronal suture fusion from *TWIST1* and *TCF12* mutations in the zebrafish and the mouse. This paper is significant as it demonstrates conservation of mechanisms underlying coronal suture fusion and presentation of Twist1/Tcf12 phenotypes in the zebrafish. The authors show independence of these suture pathologies to embryological tissue origin, suggesting that the common/ancestral developmental specification of suture formation may have different causes. The authors highlight changes in growth of the calvaria parallel/presage adult pathologies suggesting that changes in growth rate and potential may underlie observed pathologies. To support these observations the authors detail changes in the number of proliferative cells in the osteogenic fronts suggesting the defect in rate maybe attributable to lack of maintenance of progenitors/stem cells. The implications of their findings are first and foremost the conservation of mechanisms of craniosynostosis between zebrafish and mouse. This matter has been of much debate in the field with little data to support either opinion. The discussion of the similarities between the models and the independence to embryological tissue of origin (mesoderm/neural crest) is well done and important for the presentation of this work to the field. Second, the developmental analysis clearly pointed to early growth events as being primary to the changes in suture phenotypes in the models. This is in stark contrast to the center of much of the work in the field centering on suture mesenchyme and late maintenance of patency. The causes of the early shifts in proliferative capacity may be diverse, but the change in growth may be a shared phenotype underlying suture pathologies associated with synostosis.

Although these are strong points, the mechanistic data to support the changes in growth as underlying suture pathology is thin. My comments fall to two general areas:

1) Of general concern is the lack of sufficient experimental evidence to support conclusions of changes in postnatal stem cells as key finding of their mechanism. The authors look to BrdU and Sp7 expression to argue for changes in progenitor populations and proliferative potential. This measure is nicely tied to changes in osteogenic progression (surface). This is a measure of proliferation and not stem cells per se; the decrease in numbers could be in fact due to altered production of mitogens such as Fgf signaling factors that have been associated with altered suture fusion. Notably, Twist function is tied to *Fgfr2*. Further, without label retention studies, lineage tracing, or use of different progenitor markers such as *gli1* and *prrx1*, it is not appropriate to conclude that this is due to altered maintenance of postembryonic stem cell populations. There is little evidence for the effect being due to alteration in stem cells and supporting their discussion of this as the mechanism for the differential growth rate.

As one of the strengths of this work is the commonalities between mouse and zebrafish providing dual models to approach mechanism, it is imperative that more of the mechanisms be ironed out, lest the impact of the findings here highlighted above will be lessened.

2) The detailing of the growth dynamics as an early causative mechanism for how suture defects arise is interesting and potentially transformative for the field. However, the descriptions presented in this manuscript are quite descriptive and do not attempt to integrate how expression of key CS associated gene such as Fgfs are varied in their mutants, nor how alteration in Twist1/Tcf12 may affect their expression.

*Reviewer #3:*

Cranial sutures are thought to contain stem cells and this proposal has driven a lot of recent research into craniosynostosis. The suggestion that embryonic and postnatal stem cell populations are distinct is intriguing.

This study is limited to closure of the coronal suture in a specific model for craniosynostosis. Although the coronal suture is the suture that is most frequently closed prematurely, other sutures, especially of the facial skeleton have been shown to close prematurely in humans with craniosynostosis syndromes and in Fgfr-related mouse models for craniosynostosis. Have the authors looked carefully at other sutures, facial or vault? Would they expect/not expect suture closure in other sutures? Is the proposal of stem cell mechanism specific to the coronal suture? To cranial vault sutures? Or would you expect to see them in facial sutures too? Why or why not? The fact that you report other skull bone abnormalities local to the vault sutures makes me wonder if we have the whole story here. To focus only on the coronal suture limits the impact of these findings and makes it unclear why the coronal suture is targeted (given there is no mesoderm-nc boundary at this suture in fish).

The question of whether sutures are homologous across vertebrates is an interesting question. It has been proposed that the mechanism underlying the demonstrated reduction in number of bones of the modern vertebrate skull relative to the skull of pre-synapsid mammals may be similar to mechanisms underlying premature suture closure in modern humans. [True evolutionary homology of sutures would be difficult but not impossible to demonstrate across vertebrates. I am *not* asking that you do this.]

The live imaging of the sutures is a great tool providing important information.

The result that Tcf12 and Twist1 have a "conserved" early function during skull bone growth to regulate the balance of progenitors and osteoblasts – is only demonstrated for fish here, but it is a beginning to show conservation across vertebrates.

The Discussion is overly long and does not provide a solid explanation of what was observed. If there truly are no other fusions in the skulls of these fish, why does the coronal suture respond specifically to these mutations?

[Editors' note: further revisions were requested prior to acceptance, as described below.]

Thank you for resubmitting your work entitled "Altered progenitor dynamics prefigure craniosynostosis in a zebrafish model of Saethre-Chotzen syndrome" for further consideration at *eLife*. Your revised article has been favorably evaluated by Marianne Bronner as the Senior Editor, a Reviewing Editor, and the 3 original reviewers.

The manuscript has been improved but there are remaining issues that need to be dealt with. Overall, the reviewers appreciated the effort involved in generating the data in Figure 7 to strengthen the argument that this is due to loss of progenitors. However, all three reviewers had significant concerns about the quantification of this data and whether it truly does reflect loss of the progenitors. This is deemed very important since your proposed mechanism relies on this quantification. A sampling of their concerns includes:

a) "It is hard to see what the metrics of their quantitation are as I cannot see how "area" of expression can be garnered from those sections – especially when there are 4-6 foci within a section to demarcate regions. Statements of new markers for suture progenitors seems a bit overstated as the expression is generally across the bones – showing lack of expression in non-suture locations, or whole mounts might be more convincing. "

b) "I found it very hard to see the orange and magenta lines in Figure 7A and once I blew it up big enough to be able to see it, I was unconvinced by the description offered in the figure legend. According to the authors, *prxx1a* is broadly expressed at the bone fronts and in the sutures and perisosteum at 10mm and in the sutures and periosteum at adult stages. I felt more data, or better figures were required to support their overall conclusion that their findings indicate a selective exhaustion of osteoprogenitors at the developing coronal suture – as opposed to more generally across skull bones. "

c) "I still don't prefer this sentence, beginning: "As in mice and humans, coronal suture loss correlated with reduced anterior-posterior growth of the frontal and parietal bones, resulting in asymmetrically shaped heads when the suture was lost unilaterally." As written, the authors still ascribe causality of head asymmetry to suture loss, whereas both head asymmetry and unilateral suture loss may be consequences of asymmetrically reduced growth. "

Based on these concerns, we request that you either:

1) Provide more convincing quantification to ascribe this to progenitor loss.

2) If you feel that this cannot be done, then a rewording of the title and discuss to indicate that the study centers on the role of Twist1/Tcf12 in a zebrafish model of CS, and then discuss the possibility that this could be due to progenitor loss although not proven.

---

## [Author Response]

Reviewer #2:[…]1) Of general concern is the lack of sufficient experimental evidence to support conclusions of changes in postnatal stem cells as key finding of their mechanism. The authors look to BrdU and Sp7 expression to argue for changes in progenitor populations and proliferative potential. This measure is nicely tied to changes in osteogenic progression (surface). This is a measure of proliferation and not stem cells per se; the decrease in numbers could be in fact be due to altered production of mitogens such as Fgf signaling factors that have been associated with altered suture fusion. Notably, Twist function is tied to Fgfr2. Further, without label retention studies, lineage tracing, or use of different progenitor markers such as gli1 and prrx1, it is not appropriate to conclude that this is due to altered maintenance of postembryonic stem cell populations. There is little evidence for the effect being due to alteration in stem cells and supporting their discussion of this as the mechanism for the differential growth rate.As one of the strengths of this work is the commonalities between mouse and zebrafish providing dual models to approach mechanism, it is imperative that more of the mechanisms be ironed out, lest the impact of the findings here highlighted above will be lessened.

We agree that our original experiments had only inferred stem cell populations in the mutant calvarial bones and sutures through analysis of proliferation and osteoblast production. In a new Figure 7, we now directly examine progenitor populations in zebrafish using three markers (*gli1, grem1a, prrx1a*), both at a stage when the calvarial bones are still growing and at a later stage when the sutures have formed. Consistent with our original model, we find that suture progenitor cells marked by these genes are reduced in number at the developing coronal but not the sagittal suture. We also now show a similar depletion of Gli1^+^ and Grem1^+^ progenitors at the mouse coronal but not the sagittal suture at embryonic stages. As described in a new Results subsection (“Selective reduction of the osteoprogenitor pool at the mutant coronal suture”), these new experiments show molecular homology between zebrafish and mouse sutural stem cells, reveal *Grem1* as a potential new sutural stem cell marker in both species, and confirm selective exhaustion of osteoprogenitors at the coronal versus the sagittal suture.

2) The detailing of the growth dynamics as an early causative mechanism for how suture defects arise is interesting and potentially transformative for the field. However, the descriptions presented in this manuscript are quite descriptive and do not attempt to integrate how expression of key CS associated gene such as Fgfs are varied in their mutants, nor how alteration in Twist1/Tcf12 may affect their expression.

In a new Figure 7—figure supplement 1, we identify expression of *Fgfr2* and the Fgf target gene *dusp6* at the developing coronal suture of zebrafish. In mutants, we find that the domain of osteoprogenitors expressing *fgfr2* and *dusp6* is reduced, yet the remaining bone front cells express similar levels of both genes.

Subsection “Selective reduction of the osteoprogenitor pool at the mutant coronal suture”: “Thus, the calvarial phenotypes observed are unlikely to be due to whole-scale loss of Fgfr2 signaling at the bone fronts.”

Reviewer #3:Cranial sutures are thought to contain stem cells and this proposal has driven a lot of recent research into craniosynostosis. The suggestion that embryonic and postnatal stem cell populations are distinct is intriguing.This study is limited to closure of the coronal suture in a specific model for craniosynostosis. Although the coronal suture is the suture that is most frequently closed prematurely, other sutures, especially of the facial skeleton have been shown to close prematurely in humans with craniosynostosis syndromes and in Fgfr-related mouse models for craniosynostosis. Have the authors looked carefully at other sutures, facial or vault? Would they expect/not expect suture closure in other sutures? Is the proposal of stem cell mechanism specific to the coronal suture? To cranial vault sutures? Or would you expect to see them in facial sutures too? Why or why not? The fact that you report other skull bone abnormalities local to the vault sutures makes me wonder if we have the whole story here. To focus only on the coronal suture limits the impact of these findings and makes it unclear why the coronal suture is targeted (given there is no mesoderm-nc boundary at this suture in fish).*The question of whether sutures are homologous across vertebrates is an interesting question. It has been proposed that the mechanism underlying the demonstrated reduction in number of bones of the modern vertebrate skull relative to the skull of pre-synapsid mammals may be similar to mechanisms underlying premature suture closure in modern humans. [True evolutionary homology of sutures would be difficult but not impossible to demonstrate across vertebrates. I am* not asking that you do this.]The live imaging of the sutures is a great tool providing important information.The result that Tcfl2 and Twist1 have a "conserved" early function during skull bone growth to regulate the balance of progenitors and osteoblasts – is only demonstrated for fish here, but it is a beginning to show conservation across vertebrates.The Discussion is overly long and does not provide a solid explanation of what was observed. If there truly are no other fusions in the skulls of these fish, why does the coronal suture respond specifically to these mutations?

We have developed a zebrafish model for a particular form of craniosynostosis, Saethre-Chotzen Syndrome, which almost exclusively affects the coronal suture in mice and humans. A significant and unexpected finding from our study is that the coronal suture is likewise most sensitive to loss of Twist1 and Tcf12, despite this forming at a mesoderm-mesoderm boundary in fish and a neural crest-mesoderm boundary in mammals. While we have not detected loss of other sutures in our fish mutants, we do observe occasional ectopic sutures and other skull anomalies in different mutant combinations, which we describe in detail in Figure 1—figure supplement 3.

Subsection “Specific loss of the coronal suture in *tcf12; twist1b* mutant zebrafish”: “Nonetheless, our results demonstrate that, as in humans and mice with reduced *TCF12* and/or *TWIST1* dosage, mutations in the homologous genes in zebrafish result primarily in loss of the coronal suture, although other calvarial defects are also rarely observed.”

In response to Reviewer #2(see new Figure 7), we have also examined several progenitor markers (*prrx1a, grem1a, gli1*) in mutant fish and mice, which reveal selective exhaustion of parietal and frontal bone progenitor pools. We also now show altered directional growth of the parietal and frontal bones in mutants, which brings these bones together earlier than normal at the prospective coronal suture zone. Accordingly, we have extensively revised the Discussion to incorporate these new findings and their implications while condensing other sections to make it more focused.

Discussion, second paragraph: “In particular, we found that increased diagonal growth in mutants brought the parietal and frontal bones together at the prospective medial regions of the coronal suture much earlier than in wild type animals, which correlates with the medial region of the coronal suture being most commonly fused in mutants. This altered directional growth might be one reason why the coronal suture is preferentially affected in both zebrafish and mouse Twist1/Tcf12 mutants, despite the different origins of the coronal suture in these species.”

Discussion, third paragraph: “We did not observe preferential expression of *tcf12* and *twist1b* at the coronal suture in fish, consistent with similarly broad sutural expression of *Twist1* in mice (Rice et al., 2000). Instead, osteoprogenitors in the coronal zone might be fewer in number at initial stages and/or more sensitive to loss of Tcf12 and Twist1, for example due to compensation by related genes at other sutures.”

[Editors' note: further revisions were requested prior to acceptance, as described below.]

The manuscript has been improved but there are remaining issues that need to be dealt with. Overall, the reviewers appreciated the effort involved in generating the data in Figure 7 to strengthen the argument that this is due to loss of progenitors. However, all three reviewers had significant concerns about the quantification of this data and whether it truly does reflect loss of the progenitors. This is deemed very important since your proposed mechanism relies on this quantification. A sampling of their concerns includes:a) "It is hard to see what the metrics of their quantitation are as I cannot see how "area" of expression can be garnered from those sections – especially when there are 4-6 foci within a section to demarcate regions. Statements of new markers for suture progenitors seems a bit overstated as the expression is generally across the bones – showing lack of expression in non-suture locations, or whole mounts might be more convincing. "

We agree that quantification was difficult in zebrafish mutants given the small numbers of cells and low RNA signal. We have therefore removed the quantification from Figure 7, as well as the accompanying statement claiming loss of progenitors at the developing coronal suture of zebrafish.

b) "I found it very hard to see the orange and magenta lines in Figure 7A and once I blew it up big enough to be able to see it, I was unconvinced by the description offered in the figure legend. According to the authors, prxx1a is broadly expressed at the bone fronts and in the sutures and perisosteum at 10mm and in the sutures and periosteum at adult stages. I felt more data, or better figures were required to support their overall conclusion that their findings indicate a selective exhaustion of osteoprogenitors at the developing coronal suture – as opposed to more generally across skull bones. "

We now provide magnified images in Figure 7A to better visualize the developing coronal sutures in zebrafish. We also added arrows showing the periosteal expression of *prrx1a* but not *grem1a* and *gli1* within the periosteum, and arrows showing suture expression. As stated below, we no longer claim exhaustion of osteoprogenitors in zebrafish at the 10 mm stage, instead only referring to the more convincing mouse data in Figure 7C.

c) "I still don't prefer this sentence, beginning: "As in mice and humans, coronal suture loss correlated with reduced anterior-posterior growth of the frontal and parietal bones, resulting in asymmetrically shaped heads when the suture was lost unilaterally." As written, the authors still ascribe causality of head asymmetry to suture loss, whereas both head asymmetry and unilateral suture loss may be consequences of asymmetrically reduced growth. "

We agree that “resulting in” still implied causality. We therefore changed the sentence to better reflect our findings and avoid claims of causality: “…in cases where the suture was lost unilaterally we consistently observed reduced anterior-posterior growth of that side of the skull”.

Based on these concerns, we request that you either:1) Provide more convincing quantification to ascribe this to progenitor loss.2) If you feel that this cannot be done, then a rewording of the title and discuss to indicate that the study centers on the role of Twist1/Tcf12 in a zebrafish model of CS, and then discuss the possibility that this could be due to progenitor loss although not proven.

We agree that the quantification of the progenitor loss in zebrafish sutures was difficult due to the small numbers of cells and low level of expression of the genes examined. This made it hard to conclusively count “progenitors”, especially in mutants where the bones are abnormally shaped and more closely apposed. We have therefore chosen option (2) and revised the title and Discussion accordingly.

The title is changed from “*Altered progenitor dynamics* prefigure…” to “*Altered bone growth* dynamics prefigure craniosynostosis in a zebrafish model of Saethre-Chotzen syndrome”. We feel that this better reflects our in vivo imaging data in zebrafish revealing that the extent of altered bone growth predicts which mutants will develop coronal suture defects. The title also reflects our conceptual advance of developing the first genetic model of CS in zebrafish.

We have removed the quantification of “progenitor domains” at the zebrafish sutures (originally Figure 7B), and no longer refer to progenitor exhaustion at mutant zebrafish sutures. For example, in the Discussion: “Whether similar osteoprogenitors become exhausted at the developing coronal suture of zebrafish remains to be determined, as it was difficult to precisely quantify the numbers of these progenitors by RNA expression alone.”

When referring to reduced progenitors at the mutant coronal suture, we now only refer to the mouse data in Figure 7C which we feel is convincing – the reviewers only expressed concern with the zebrafish data in Figure 7A. For example, in the Discussion: “These findings … suggest, at least in mice, a selective exhaustion of osteoprogenitors at the developing coronal suture”.

Rather than stating that our results prove progenitor exhaustion as the mechanism underlying coronal suture loss, we have altered the language throughout to present progenitor exhaustion as one plausible model consistent with our data. For example, in the Abstract: “These findings … suggest that the coronal suture might be especially susceptible to imbalances in progenitor maintenance and osteoblast differentiation.”

We have also added new language that Gli1, Grem1, and Prrx1 label “putative” progenitors in the developing sutures, given that lineage tracing studies at these stages would still be needed to confirm this.